# Engineering monocyte/macrophage—specific glucocerebrosidase expression in human hematopoietic stem cells using genome editing

Samantha G. Scharenberg [1,4], Edina Poletto [2,4], Katherine L. Lucot [3], Pasqualina Colella [1], Adam Sheikali[1], Thomas J. Montine[3], Matthew H. Porteus [1✉] & Natalia Gomez-Ospina [1✉]

Gaucher disease is a lysosomal storage disorder caused by insufficient glucocerebroside activity. Its hallmark manifestations are attributed to infiltration and inflammation by macrophages. Current therapies for Gaucher disease include life—long intravenous administration of recombinant glucocerebroside and orally-available glucosylceramide synthase inhibitors. An alternative approach is to engineer the patient's own hematopoietic system to restore glucocerebrosidase expression, thereby replacing the affected cells, and constituting a potential one-time therapy for this disease. Here, we report an efficient CRISPR/Cas9-based approach that targets glucocerebrosidase expression cassettes with a monocyte/macrophage-specific element to the CCR5 safe-harbor locus in human hematopoietic stem and progenitor cells. The targeted cells generate glucocerebroside-expressing macrophages and maintain long-term repopulation and multi-lineage differentiation potential with serial transplantation. The combination of a safe-harbor and a lineage-specific promoter establishes a universal correction strategy and circumvents potential toxicity of ectopic glucocerebrosidase in the stem cells. Furthermore, it constitutes an adaptable platform for other lysosomal enzyme deficiencies.

[1] Department of Pediatrics, Stanford University School of Medicine, Stanford, CA, USA. [2] Gene Therapy Center, Hospital de Clinicas de Porto Alegre, Porto Alegre, Brazil. [3] Department of Pathology, Stanford University School of Medicine, Stanford, CA, USA. [4]These authors contributed equally: Samantha G. Scharenberg, Edina Poletto. ✉email: mporteus@stanford.edu; gomezosp@stanford.edu

Gaucher disease (GD) is genetic disorder caused by mutations in the *GBA* gene that result in glucocerebrosidase (GCase) deficiency and the accumulation of glycolipids in cell types with high-glycolipid degradation burden, especially macrophages[1]. GD encompasses a spectrum of clinical findings from a perinatal-lethal form to mildly symptomatic forms. Three major clinical types delineated by the presence (types 2 and 3) or absence (type 1) of central nervous system involvement are commonly used for determining prognosis and management[2]. In western countries, GD type 1 (GD1) is the most common phenotype (~94% of patients) and typically manifests with hepatosplenomegaly, bone disease, cytopenias, and variably with pulmonary disease, as well as elevated risk for malignancies and Parkinson's disease[3,4].

The pathophysiology in GD1 is thought to be driven by glucocerebroside-engorged macrophages that infiltrate the bone marrow, spleen and liver, and promote chronic inflammation, as well as low-grade activation of coagulation and complement cascades[5–7]. Current therapies for GD1 include orally available small-molecule inhibitors of glucosylceramide synthase (substrate reduction therapy or SRT) and glucocerebrosidase enzyme replacement (ERT) targeted to macrophages via mannose receptor-mediated uptake[8]. While ameliorative for visceral and skeletal disease manifestations, these therapies are chronically administered, life-long, and costly. Allogeneic hematopoietic stem-cell transplantation (allo-HSCT) has been applied successfully as a one-time treatment for GD1[9] and its therapeutic effect is achieved by supplying graft-derived GCase-competent macrophages. However, because of the significant transplant-related morbidity and mortality of allo-HSCT, ERT, and SRT are standard of care for patients with GD1[10,11].

The effectiveness of macrophage-targeted ERT and allo-HSCT for treating GD1 suggests that restoration of GCase function in macrophages alone is sufficient for phenotypic correction in GD1. Consequently, restoring GCase activity in the patient's own hematopoietic system to establish an autologous approach that averts many of the risks of allo-HSCT could be a safer and potentially curative therapy for this disease. Furthermore, unlike ERT and the best tolerated SRT, it could provide enzyme reconstitution in the brain that could benefit neuronopathic forms of the disease[9]. For these reasons, non-targeted gene addition into human hematopoietic stem and progenitor cells (HSPCs) have been explored, first using retroviruses[12–15] and later lentiviral vectors, and have yielded promising results in murine GD models[16–18]. Nevertheless, concerns remain about the potential for insertional mutagenesis and malignant transformation in viral gene transfer[19,20] stressing the need for the development of targeted gene addition strategies to generate genetically modified HSPCs for human therapy.

Modern genome-editing tools can achieve genetic modifications and integrations with single-base pair precision[21]. A highly engineerable platform derived from the bacterial CRISPR/Cas9 system has been optimized for gene editing in HSPCs[22–24]. This platform consists of two main components: (1) a sgRNA/Cas9 ribonucleoprotein complex (RNP) functioning as an RNA-guided endonuclease, and (2) a designed homologous repair template delivered using adeno-associated viral vector serotype six (AAV6). The RNP comprises a 100-bp, chemically modified, synthetically generated, single-guide RNA (sgRNA) complexed with *Streptococcus pyogenes* Cas9-endonuclease and delivered into the cells by electroporation[25]. In the nucleus, the RNP binds to the target sequence and Cas9 catalyzes a double-stranded break, stimulating one of two repair pathways: (1) non-homologous end joining (NHEJ), in which broken ends are directly ligated, often producing small insertions and deletions (indels); and (2) homology-directed repair (HDR), in which recombination with the supplied homologous repair template is used for precise sequence changes[21]. In human HSPCs, the AAV6 genome is an efficient delivery method for the homologous repair templates containing an experimenter-defined genetic change flanked by homology arms centered at the break site[22]. Accordingly, the HDR pathway can be leveraged not only to achieve single-base pair changes, but also to integrate entire expression cassettes into a non-essential safe harbor locus, thus enabling stable expression of tailorable combinations of regulatory regions, transgenes, and selectable markers[24,26]. One potential safe harbor locus is *CCR5*. This gene encodes the major co-receptor for HIV-1, and is considered a non-essential locus because of the high prevalence of healthy homozygous *CCR5*$^{\Delta 32}$ individuals in European populations (>10%)[27] and the observation that homozygous carriers of the Δ32 mutation are resistant to HIV-1 infection[28].

Here, we describe our generation and characterization of GCase-targeted human HSPCs, a crucial step towards establishing autologous transplantation of genome-edited cells for GD. We use the RNP/AAV6 platform to achieve efficient integration of GCase cassettes into the CCR5 safe harbor locus. By leveraging a lineage-specific promoter highly expressed in the monocyte/macrophage lineage, we achieve GCase expression in the affected cell lineages while also minimizing ectopic expression in hematopoietic stem and progenitor compartments. GCase-targeted HSPCs demonstrate the capacity for long-term engraftment and multi-lineage differentiation, including the generation of functional macrophages with supraphysiologic GCase expression in vivo.

## Results

**Efficient targeting of GCase to the *CCR5* locus in human HSPCs.** We used the CRISPR/Cas9 and AAV system to target glucocerebrosidase (GCase) expression cassettes to the human *CCR5* safe harbor locus (Fig. 1a). The sgRNA targeting the third exon of *CCR5* was previously validated for high on-target activity in primary human HSPCs[24,29] and has excellent specificity as prior studies failed to reveal any detectable off-target activity using high-fidelity Cas9[24]. AAV donor repair templates were generated to drive GCase expression by two different promoters: (1) the Spleen Focus-Forming Virus (SFFV) promoter, which drives constitutive supraphysiologic expression; and (2) the CD68S promoter, a shortened derivative of the endogenous human CD68 promoter with expression restricted to the monocyte/macrophage lineage[30,31] (Fig. 1b). This lineage-specific promoter was chosen to minimize potential complications of GCase overexpression in the stem-cell compartment. The Citrine-containing vectors were designated SFFV-GCase-P2A-Citrine and CD68S-GCase-P2A-Citrine. A third AAV, CD68S-GCase, lacking the reporter protein, was developed as a more clinically relevant vector for in vivo studies (Fig. 1a).

The targeting efficiencies achievable for each vector were determined by the percent of Citrine-positive (Citrine+) cells and by the percent of *CCR5* alleles with on-target cassette integrations using molecular analysis (giving the cell and allele targeting frequencies, respectively). In the presence of both AAV and RNP, the SFFV-driven cassette resulted in approximately $51.5 \pm 9.1\%$ (mean ± SD) Citrine+ HSPCs 48-h post-targeting, while AAV alone produced $5.9 \pm 4.2\%$ dim Citrine+ cells, likely reflecting episomal expression (Fig. 1c, d). The fraction of *CCR5* alleles with on-target cassette integration in the unselected population was $29 \pm 9\%$ as measured by droplet digital PCR (ddPCR) (Fig. 1e and Supplementary Fig. 1a). To verify targeting in Citrine+ cells, these cells were sorted by FACS and the fraction of modified alleles measured (Fig. 1e and Supplementary Fig. 1a). The allelic modification frequency of HSPCs treated with the SFFV-GCase-P2A-Citrine vector that were Citrine+ (SFFV-GCase-Citrine+) was $65.9 \pm 4.9\%$,

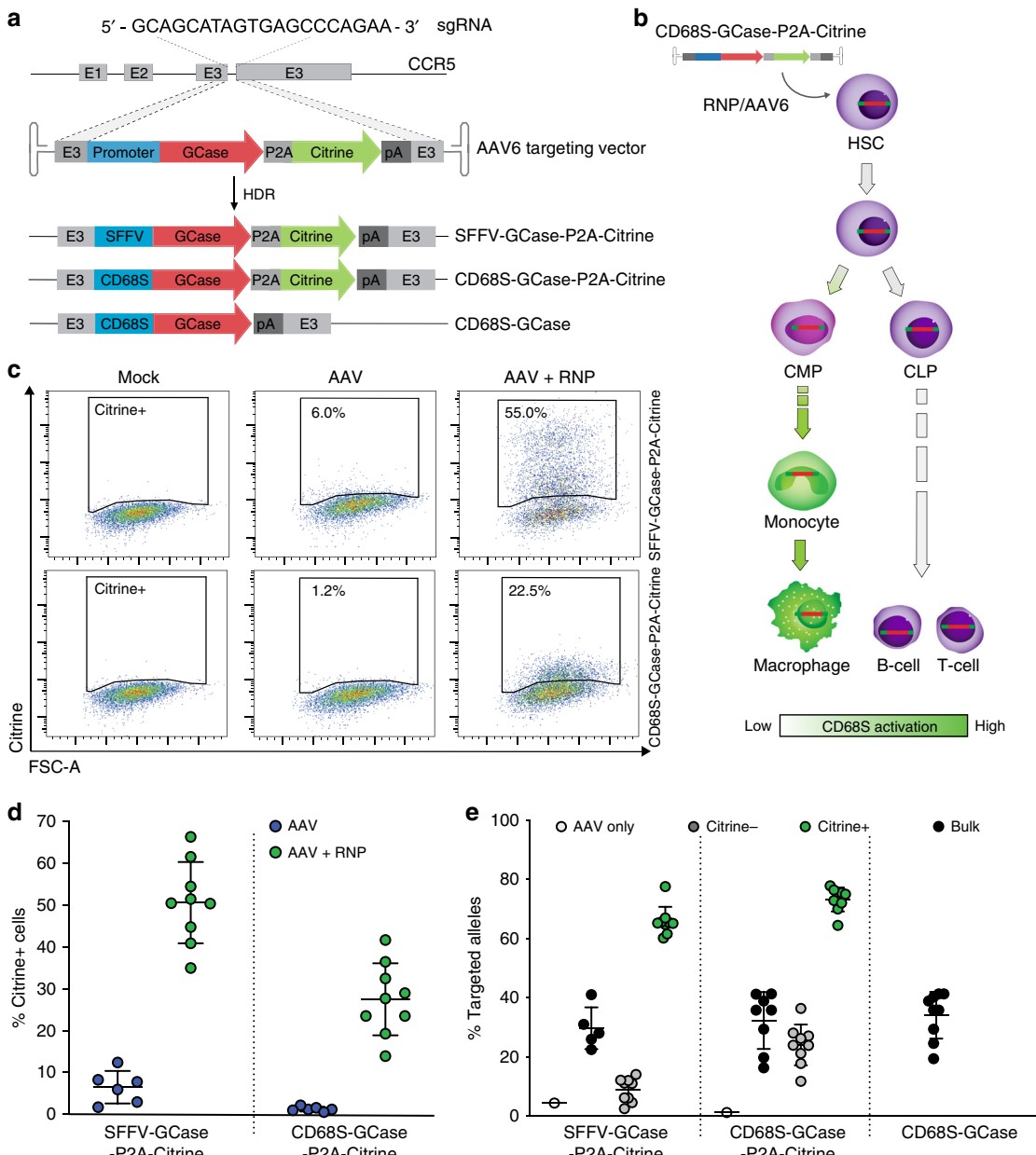

**Fig. 1 Efficient targeting of GCase to the CCR5 locus in human HSPCs 48-hours post-modification. a** Schematic of gene targeting mediated by sgRNA/ Cas9 RNP and rAAV targeting vectors where E1-3 are *CCR5* exons. **b** Schematic of expected CD68S promoter activity. Green indicates activation. **c** Representative flow plots of Citrine expression versus forward scatter (FSC) for HSPCs without treatment (mock), treated with rAAV alone (AAV), and treated with RNP and rAAV (RNP+AAV). **d** Flow cytometric quantification of Citrine+ HSPCs targeted with SFFV-GCase-P2A-Citrine and CD68S-GCase-P2A-Citrine vectors in the presence (green circles) or absence (blue circles) of RNP (*n* = 9 biologically independent human donor samples). **e** Percent of *CCR5* alleles with integrated CD68S-GBA-P2A-Citrine and SFFV-GBA-P2A-Citrine cassettes in AAV only (white), bulk (black), FACS-enriched Citrine− (gray) and Citrine+ (green) HSPCs, and in bulk CD68S-GCase-targeted cells (black). Data shown as mean ± SD. Source data are provided as a Source Data file.

corresponding to 69% and 31% mono-allelically and bi-allelically targeted cells, respectively. Genotyping of single-cell-derived colonies corroborated that 98% percent of the Citrine+ HSPCs were targeted and, consistent with the ddPCR data, showed 67% mono-allelic and 33% bi-allelic targeting (Supplementary Fig. 1b–d).

We predicted that because the CD68S promoter should be lineage-specific, Citrine would not be highly expressed in stem and non-myeloid biased progenitor cells and therefore, Citrine expression in HSPCs would not reflect the true editing efficiency of the CD68S-P2A-GCase-Citrine vector (Fig. 1b). Consistent with this, we found that at 48-h post-modification, Citrine

expression from HSPCs treated with the CD68S-GCase-P2A-Citrine AAV and RNP was dim (mean fluorescence intensity (MFI) was 24-fold lower than for the SFFV-GCase- Citrine+ cells) and the mean percentage of CD68S-GCase-Citrine+ HSPCs was 27.7 ± 8.5%, significantly lower than for the SSFV-driven construct despite having comparable *CCR5* allele targeting frequencies (32.3 ± 9.6%) (Fig. 1c–e). Most importantly, the allele targeting frequency within the CD68S-GCase-Citrine-negative population (CD68S-GCase-Citrine−) ranged from 11.8 to 36.4%, confirming the presence of targeted cells lacking Citrine expression (Fig. 1e). We reasoned that the subset of

CD68S-GCase-Citrine+ HSPCs likely comprise a subpopulation of granulocyte-monocyte-committed progenitors with increased CD68S promoter activation, while CD68S-GCase-Citrine− HSPCs contain the more primitive populations. Single-cell-derived colony genotyping confirmed that 96.5% of the Citrine+ cells had targeted cassette integrations and showed frequencies of mono-allelic and bi-allelic editing of 64% and 36%, respectively (Supplementary Fig. 1d). The allele targeting frequency of the CD68S-GCase vector lacking Citrine was 35.8 ± 7.9% in unselected cell populations corresponding to ~52% of cells having targeted integrations (Fig. 1e).

**Generation of human GCase-macrophages from edited HSPCs**. One mechanism by which HSCT is therapeutic in Gaucher disease is through the generation of GCase-expressing macrophages. To confirm the development of macrophages from GCase-targeted HSPCs, we first differentiated control human CD34+ HSPCs using a cytokine cocktail, including M-CSF, GM-CSF, SCF, IL-3, FLT3 ligand, and IL-6[32]. HSPCs differentiated in this manner exhibited characteristic ameboid morphology as well as expression of the monocyte/macrophage lineage markers CD14 and CD11b, with concurrent loss of the HSPC marker CD34 (Fig. 2a, b and Supplementary Fig. 2a). Following the same differentiation protocol, human HSPCs targeted with the SFFV-GCase-P2A-Citrine and CD68S-GCase-P2A-Citrine constructs, produced macrophages that exhibited Citrine expression, characteristic morphology, and normal phagocytosis of pHrodo-labeled *E. coli* (Fig. 2c). CD14 and CD11b marker expression in mock-treated, Citrine+ and Citrine− populations from these two constructs revealed comparable expression compared to unmodified cells in all conditions except in CD68S-GCase-Citrine+ cells, which had higher expression in both the standard HSPC and macrophage differentiation conditions (Fig. 2d, e and Supplementary Fig. 2b). These results indicate that GCase-targeted HSPCs can produce functional macrophages in vitro and suggest that CD68S-GCase-Citrine+ HSPCs are already primed for differentiation along this lineage.

CCR5 is absent from HSPCs but becomes expressed with monocyte/macrophage differentiation. To examine the effect of our genome editing process on CCR5 expression we targeted human HSPCs, differentiated them, and quantified CCR5 protein by FACS (Supplementary Fig. 3). In the RNP alone condition, the efficiency of double-strand DNA break generation by our CCR5 RNP complex was estimated by measuring the frequency of insertions/deletions (Indel) at the predicted cut site. The mean indel frequencies in the undifferentiated and differentiated populations was 96.8% ± 1.2 and 96.4% ± 1.6, respectively, resulting in almost complete knock-down of CCR5 protein expression (Supplementary Fig. 3a). In the presence of both RNP and AAV, cells that successfully underwent HDR (Citrine+) lacked CCR5 expression, consistent with disruption of both CCR5 alleles by either bi-allelic integration of the cassette or mono-allelic with indel formation in the second allele (Supplementary Fig. 3b). In the presence of AAV, CCR5+ cells can be found in the population that did not undergo HDR (~20%), suggesting that AAV transduction decreases indel generation or exerts a small-negative selection in cells containing both AAV and RNP.

**CD68S confines expression to the monocyte/macrophage lineage**. The CD68S cassettes were designed to selectively express GCase in the monocyte/macrophage lineage in order to prevent potential toxicity to stem cells from ectopic GCase over-expression. To validate the lineage specificity of the CD68S promoter, CD68S-GCase-Citrine+ and SFFV-GCase-Citrine+ HSPCs were cultured with growth factors that promoted either

HSPC maintenance (HSPC) or macrophage differentiation (MΦ) and Citrine expression was monitored for 20 days. As expected for a constitutive promoter, the fraction of SFFV-GCase-Citrine+ cells remained stable over time in both HSPC and MΦ cultures (>95%). An average of 9.2% and 16.3% of SFFV-GCase-Citrine− cells became positive in the HSPC and MΦ cultures, respectively, which was consistent with the presence of targeted *CCR5* alleles in this population based on ddPCR (Fig. 3a, b). When cultured long-term, the MFI of SFFV-GCase-Citrine+ cells decreased, but the drop in fluorescence intensity was seen exclusively in a subset of cells with very high Citrine expression (Supplementary Fig. 4a, b). Notably, the allele modification frequency did not differ throughout the culturing process, suggesting that the change in Citrine expression was due to regulation of transcription from SFFV promoter or translation but not to selection against the modified cells (Supplementary Fig. 4c). In contrast, the percentage of CD68S-GCase-Citrine+ cells decreased in the HSPC cultures but was maintained in the MΦ cultures (Fig. 3a, b). Moreover, there was a substantial increase (~30-fold) in Citrine MFI from CD68S-GCase-Citrine+ cells in the MΦ compared to the HSPCs culture over the 21-day differentiation (Fig. 3c).

As Citrine is only a proxy for GCase cassette expression, we also examined GCase protein expression directly by quantifying its enzymatic activity in HSPC and MΦ culture conditions. In HSPC cultures, SFFV-GCase-Citrine+ and CD68S-GCase-Citrine+ cells showed ~7.7 and 1.3-fold more GCase activity, respectively, compared to unmodified cells (mock-treated). The CD68S-GCase-Citrine− population showed the same activity as unmodified cells (1.0-fold) supporting the idea that there is no leakage GCase expression from the CD68S promoter in more primitive and non-myeloid HSPCs (Fig. 3d). Macrophages derived from CD68S-GCase-Citrine+ and SFFV-GCase-Citrine+ HSPCs expressed ~2-fold higher GCase than macrophages derived from mock-treated cells (Fig. 3e). In all but the SFFV-GCase-Citrine+ population, macrophage differentiation resulted in higher levels of GCase expression. This explains the decrease in fold expression in cells targeted with the SFFV-driven cassette with differentiation (from 7.7 to 2.3), as it reflects the marked increase in endogenous GCase (~4-fold) in the mock cells without a proportional change in exogenous GCase expression from the SFFV expression cassette (Supplementary Fig. 4d).

To examine the possibility that differential expression of the GCase cassette was due to changes in the targeted cell populations, we measured the allele targeting frequencies at the time of sorting and post-culture in the HSPC and MΦ cultures using ddPCR (Fig. 3f). We found that the percentage of alleles with on-target cassette integration within Citrine+ and Citrine− populations targeted with both cassettes did not differ between culturing conditions, thus confirming that the changes in expression were attributable to the lineage-specific activity of the CD68S promoter.

**GCase-targeted HSPCs sustain long-term hematopoiesis**. To examine the potential of GCase-HSPCs to become a one-time therapy for GD1, we tested their long-term repopulation capacity. We first assessed the colony-forming ability of the targeted HSPCs in vitro using the colony-forming unit (CFU) assay. We sorted mock, Citrine+ and Citrine− from SFFV and CD68S targeted populations as single cells in 96-well plates 48-h post-transplantation and assessed their phenotype 14 days later. Notably, SFFV-GCase-Citrine+ HSPCs produced the fewest colonies of all conditions and exhibited the highest variability in the distribution of colony phenotypes formed, suggesting that supraphysiologic GCase expression or other aspects of SFFV promoter physiology may have a toxic effect on HSPCs (Fig. 4a). As predicted by the model of restricted lineage expression of the

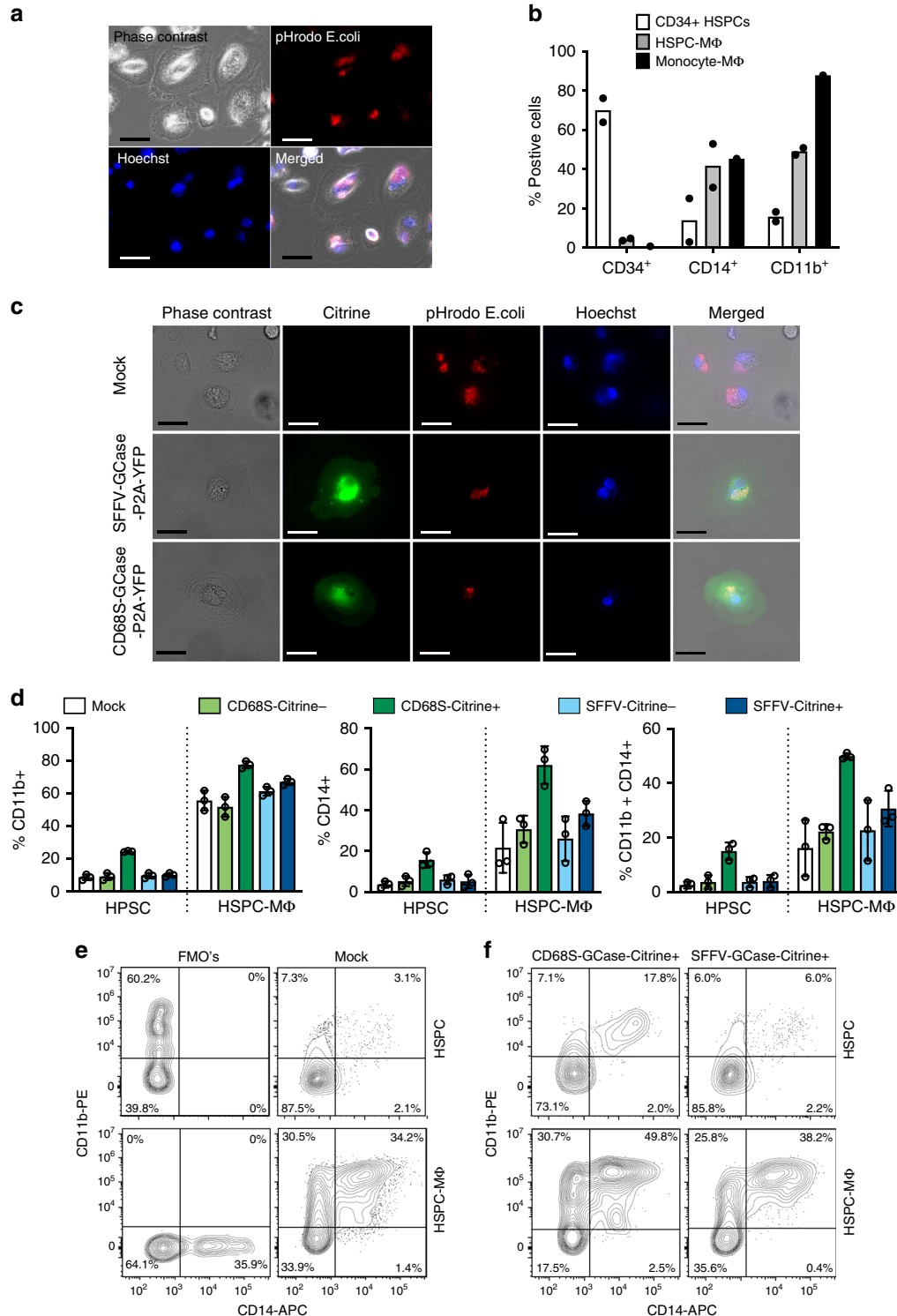

CD68S promoter, CD68S-GCase-Citrine+ HSPCs formed exclusively CFU-GM's (granulocyte/monocyte), while the cells that did not express Citrine (CD68S-GCase-Citrine−) produced a normal distribution of colony phenotypes (Fig. 4b). These results strongly support our earlier hypothesis that CD68S-GCase-Citrine+ cells in undifferentiated HSPCs represent granulocyte/monocyte primed progenitors and that bona fide CD68S-GCase–P2A-Citrine-targeted stem cells reside within the CD68S-GCase-Citrine− population.

To test in vivo engraftment potential, GCase-targeted HSPCs were serially transplanted into NOD-scid IL2Rgamma

(NSG) mice. Cell doses varied from $2.5 \times 10^5$ to $2 \times 10^6$ HSPCs and were dependent on the CD34+ cell yield per human donor. We focused our long-term engraftment experiments on the CD68S-GCase-P2A-Citrine and CD68S-GCase vectors because of the potential detrimental effect of the SFFV promoter, its observed drop in expression, and its barriers to clinical translation. Targeted cells were transplanted without selection intrafemorally or intrahepaticaly into sublethally irradiated NSG mice. Primary human engraftment was quantified after 16 weeks as the percentage of cells expressing human CD45 within the total hematopoietic

**Fig. 2 Generation of human GCase-macrophages from genome-edited HSPCs. a** Representative images showing phase contrast, phagosomes visualized by pHrodo-labeled *E.coli* (red), and nuclei (blue) in mock-treated human HSPCs after 20 days in macrophage differentiation media for one of the two samples analyzed in **b**. Scale bar 10 μm. **b** Human CD34, CD14, and CD11b marker expression in HSPC-derived macrophages (HSPC-MΦ) and human monocyte-derived macrophages (Monocyte- MΦ) after in vitro differentiation compared to undifferentiated cells (CD34+ HSPCs) (n = 2 biologically independent human donor samples). **c** Representative images showing phase contrast, Citrine expression (green), phagosomes visualized by pHrodo-labeled *E.coli* (red), and nuclei (blue) in mock-treated, SFFV-GCase-P2A-Citrine, and CD68S-GCase-P2A-Citrine targeted macrophages for one of the three samples analyzed in **d**. Scale bar 20 μm. **d** Human CD14, and CD11b marker expression in mock-treated (white), CD68S-GCase-P2A-Citrine targeted (Citrine–: light green; Citrine+: dark green), and SFFV-GCase-P2A-Citrine targeted cells (Citrine–: light blue; Citrine+: dark blue) with and without macrophage differentiation. Left graph: CD11b+. Middle graph: CD14+. Right graph: CD11b+/CD14+ (n = 3 biologically independent human donor samples). **e** Representative FACS plots of Fluorescence Minus One controls (FMO's) and Mock samples showing CD11b and CD14 expression in HSPC maintenance or Macrophage differentiation media. **f** Representative FACS plots showing CD11b and CD14 expression in CD68S-GCase-Citrine+ and SFFV-GCase-Citrine+ cells in HSPC maintenance or macrophage differentiation media. Data shown as mean ± SD. Source data are provided as a Source Data file.

population (mouse CD45+ and human CD45+, Supplementary Fig. 5).

Transplantation of GCase-targeted HSPCs resulted in substantial human cell chimerism. In the bone marrow, the median human cell chimerism was 23.2% (min: 0.17%; max: 91.5%) and 50.6% (0.53%; 91.7%) in CD68S-GCase-targeted and CD68S-GCase-P2A-Citrine-targeted cells, respectively (Fig. 4c). Similar engraftment numbers were seen in the spleen: 20.4% (0.14%; 79.3%) for the cassette lacking Citrine and 35.8% (0.38%; 89.6%) for the cassette having Citrine (Fig. 4d). To determine the proportion of engrafted cells derived from targeted HSPCs, the targeted allele frequency of the engrafted hCD45+ population in the bone marrow was measured using ddPCR in cell preparations that included mouse and human CD45+ cells as the ddPCR assay recognizes only human alleles (Fig. 4e and Supplementary Fig. 6a). The median allele targeting frequencies of the engrafted cell populations were 4.4% (min: 0.23%; max: 51.0%) and 4.2% (0.73%; 34.6%) for the CD68S-GCase and CD68S-GCase-P2A-Citrine cassettes, respectively; however, allele targeting frequency varied highly across human cell donors and mice. The allele targeting frequency of the engrafted cells tended to be lower compared to the transplanted HSPCs, with an observed drop ranging from 1.9 to 12.5-fold (Supplementary Fig. 6b). As cell doses of transplantation varied in the mice targeted with the Citrine-containing construct, the mice were colored-coded and tracked for engraftment and targeting efficiency in engrafted cells. This suggested a correlation between higher cell dose and higher engraftment of modified cells, a finding that is not surprising as there are likely more targeted long-term stem cells available for engraftment.

Serial engraftment studies are the gold standard to determine self-renewal capacity of hematopoietic stem cells. Secondary transplants were performed by isolating human CD34+ cells from bone marrow in eight 16-week mice (seven from CD68S-GCase and one from CD68S-GCase-P2A-Citrine targeted cells) and transplanting them (without pooling) into eight NSG recipient mice. Human engraftment and allele targeting frequency were assessed 16 weeks later (32 weeks post-modification) as previously described (Supplementary Fig. 7). The median human cell chimerism of all transplants was 10% (Range: 0.04%–48.9%) (Fig. 4f). Droplet digital PCR analysis of the engrafted cells from mice with human cell chimerism >1% (n = 5) showed a median allele targeting frequency of 21.9% (min: 1.3%; max: 40.5%), compared to 6.3% in the cells prior to transplantation (Fig. 4g). We reason that this increase in allelic targeting pre-to-post transplantation in secondary transplants reflects that targeted HSPCs that undergo primary engraftment in an NSG recipient have high engraftment potential and confirms the presence of long-term repopulating hematopoietic stem cells in the genome-edited population that are capable of long-term engraftment in vivo.

**In vivo differentiation of GCase-targeted HSPCs**. To examine the multi-lineage differentiation potential of GCase-targeted HSPCs in vivo we measured lymphoid and myeloid engraftment by the expression of the cell surface markers hCD19 (B-cells) and hCD33 (pan-myeloid), respectively. We included only mice with human engraftment >1% as these have sufficient cell numbers to reliably measure myeloid and lymphoid reconstitution. In primary engraftment studies, the median percentage of myeloid cells and B-cells in the bone marrow was 27.4% and 65.9%, respectively, for the mice transplanted with CD68S-GCase-targeted HSPCs, and 19.3% and 70%, respectively, for the mice transplanted with CD68S-GCase-P2A-Citrine-targeted HSPCs (Fig. 5a). In general, B-cell production was higher than myeloid and consistent with what has been previously reported for unmodified cells[33,34]. We similarly found myeloid and lymphoid cell production in secondary engraftment mice in five of the eight mice with bone marrow chimerism >1% (Fig. 5b). Mice with low human cell chimerism (<1%), have low cells numbers making the quantitation of targeted human alleles and human subpopulations less reliable.

To assess the lineage specificity of the CD68S promoter in vivo, we compared Citrine expression in the B-lymphoid and myeloid compartments in primary engraftments studies of CD68S-GCase-P2A-Citrine-targeted HSPCs that had robust engraftment of targeted cells (allele modification fraction >10%). As expected, expression of the CD68S-GBA-P2A-Citrine cassette was restricted to the myeloid (CD33+) and monocyte lineages (CD14+), with more frequent expression seen in monocytes (Fig. 5c, d). Despite robust modification in the bone marrow, three mice did not show Citrine expression in monocytes, which could be due to incomplete differentiation along this lineage since the human cells are lacking the appropriate cytokines or expression that is below our rigorous gating strategy. As the generation of GCase-expressing macrophages is critical to addressing Gaucher disease pathophysiology, it was also important to verify that engrafted, GCase-targeted HSPCs have the capacity to produce human macrophages with heterologous GCase expression. Towards this end, human CD14+ monocytes were isolated via FACS from the bone marrow of transplanted mice 16 weeks post-transplantation and differentiated by adding human macrophage colony stimulating factor (M-CSF). This step was performed in vitro because mouse M-CSF, a cytokine required for macrophage differentiation, does not have activity on human cells[35]. Human macrophages differentiated in this manner showed expression of the lineage marker CD68, as well as Citrine (12.3 ± 4.5% of human CD68+ cells), verifying that engrafted, targeted HSPCs can produce macrophages that express the therapeutic GCase cassette (Fig. 5e and Supplementary Fig. 8).

To improve engraftment and differentiation of myeloid lineages of our modified HSPCs in vivo, we performed transplantation experiments in NSG-SGM3 mice. These are

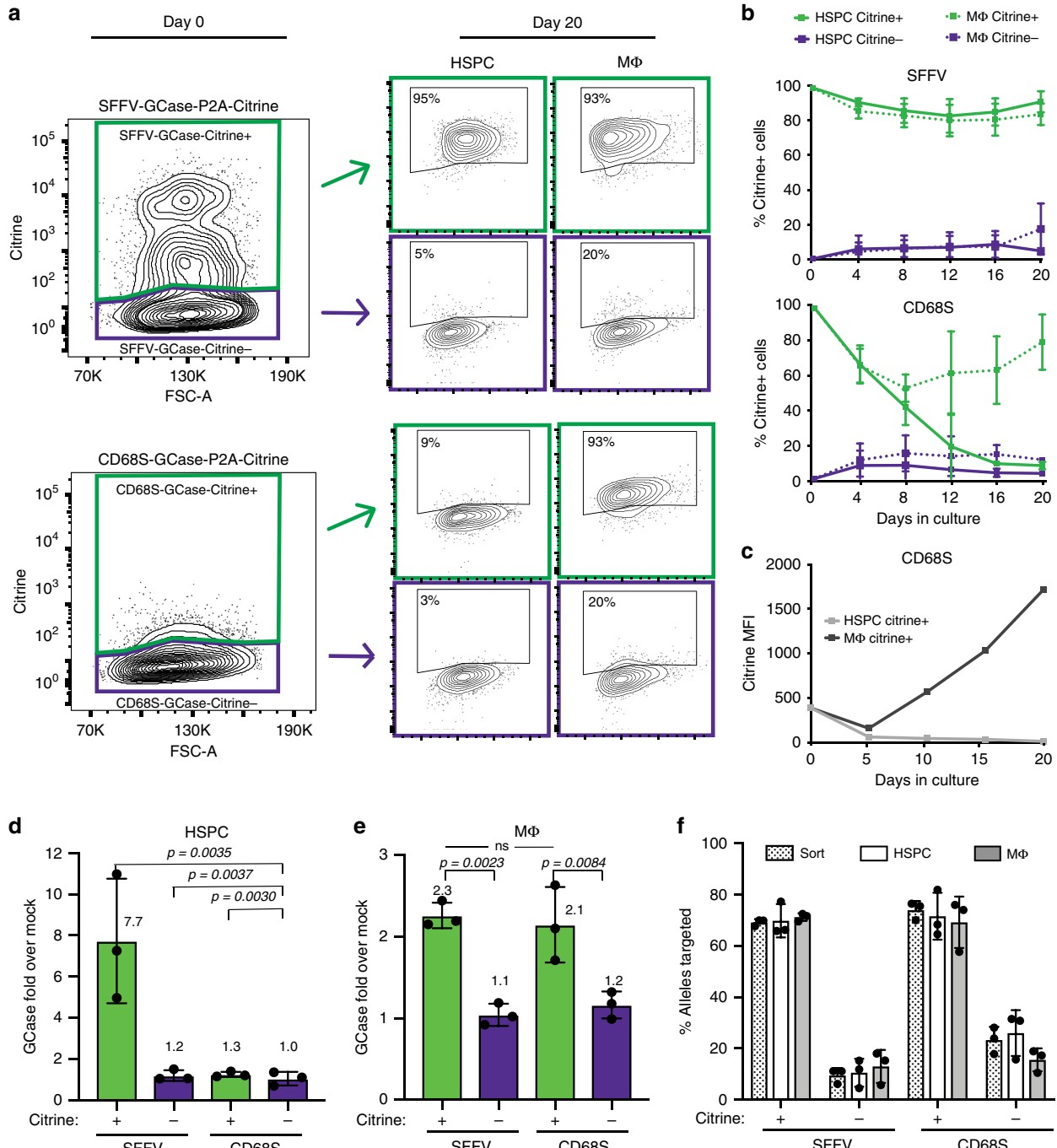

**Fig. 3 CD68S promoter confines GCase expression to the monocyte/macrophage lineage. a** Representative flow plots showing Citrine+ and Citrine− populations at the time of sort (day 0, 48-h post-modification) and after 20 days in HSPC maintenance (HSPC) or macrophage differentiation (MΦ) cultures. **b** Citrine expression expressed as %Citrine+ cells over time in HSPC and MΦ cultures. HSPC Citrine+ (solid green line), HSPC Citrine− (solid purple line), MΦ Citrine+ (dotted green line), MΦ Citrine− (dotted purple line) (n = 3 biologically independent samples). **c** Representative Citrine expression expressed MFI over time in HSPC (gray) and MΦ (black) cultures in the CD68S-GCase-P2A-Citrine-targeted cells. **d** Fold GCase activity in HSPC and **e** MΦ cultures in targeted cells compared to unmodified (mock-treated) cells (n = 3 biologically independent samples). Comparisons between groups were performed using one-way ANOVA test and post-hoc comparisons were made with the Tukey's multiple comparisons test. **f** Percent of targeted *CCR5* alleles at the time of sort (dotted) and after 20 days in HSPC (white) and MΦ (gray) cultures (n = 3 biologically independent samples). Data shown as mean ± SD. Source data are provided as a Source Data file.

NSG mice expressing human interleukin-3 (IL-3), human granulocyte/macrophage-stimulating factor (GM-CSF), and human Stem Cell Factor (SCF or KIT-ligand), cytokines that support the engraftment and differentiation of human-myeloid lineages[36,37]. At 16 weeks, transplantation of CD68S-GCase-P2A-Citrine-targeted cells resulted in median human cell chimerism of

17.7% (min: 5.1%; max: 39.6%), 61.7% (min: 22.1%; max: 85.8%), and 33.6% (min: 1.8%; max: 72%) in the bone marrow, spleen and peripheral blood, respectively (Fig. 6a). The median allele targeting frequencies of the engrafted cell populations were 15.6% (min: 12%; max: 20%), 20.4% (min: 16%; max: 25%), 5.0% (min: 2%; max: 29%) in the same tissues (Fig. 6b). The observed

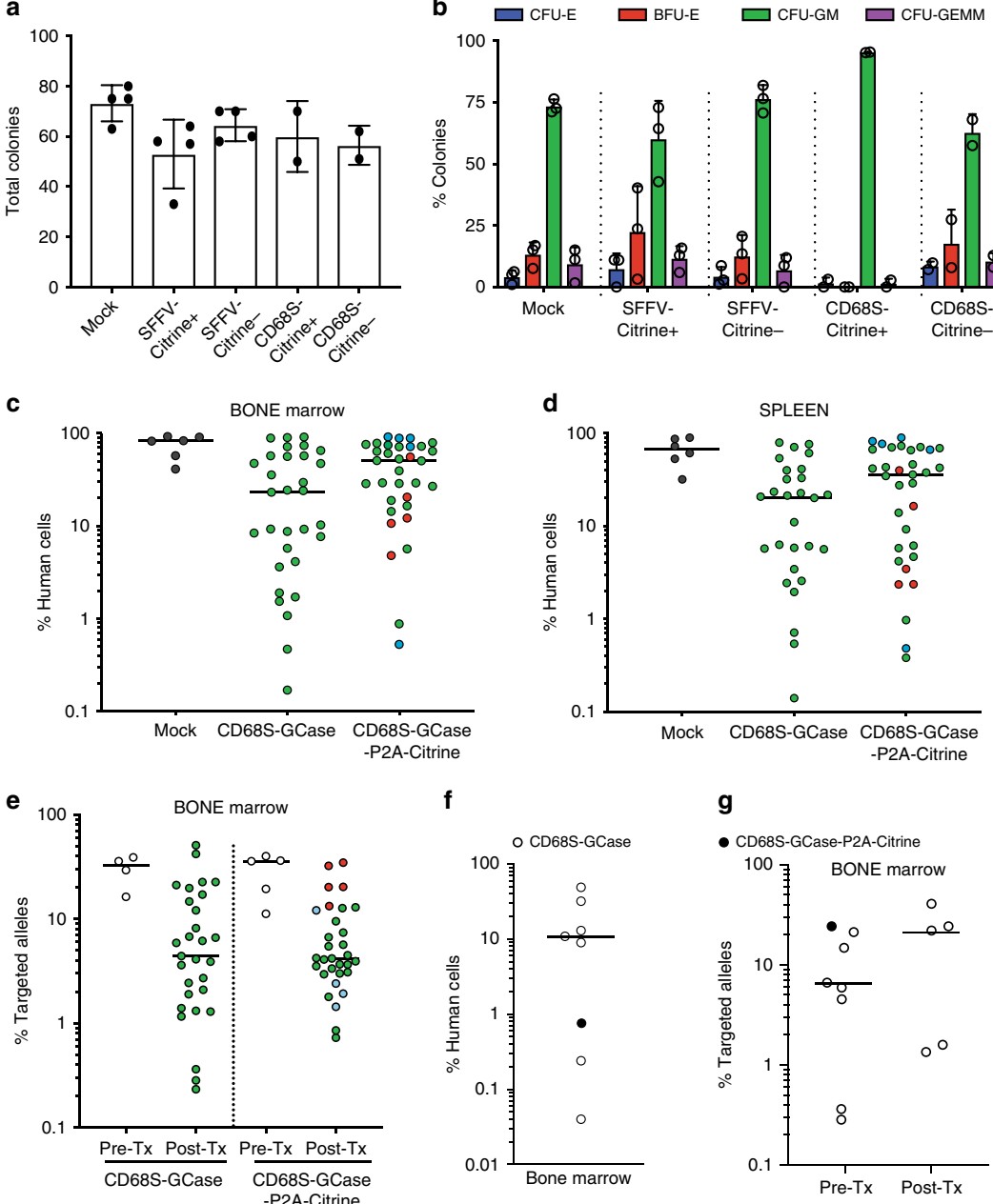

**Fig. 4 GCase-targeted HSPCs sustain long-term hematopoiesis. a** Total number of colonies formed from mock, Citrine+ and Citrine− SFFV and CD68S-driven constructs (n = 4 biologically independent human donor samples for Mock and SFFV and n = 2 for CD68S). **b** Distribution of phenotypes of colonies formed. Erythroid progenitors (burst-forming unit-erythroid or BFU-E (red)) and colony-forming unit-erythroid or CFU-E (blue), granulocyte-macrophage progenitors (CFU-GM, green), and multi-potential granulocyte, erythroid, macrophage, megakaryocyte progenitor cells (CFU-GEMM, purple) (n = 4 biologically independent human donor samples for Mock and SFFV and n = 2 for CD68S). **c** Primary human engraftment (16 weeks) in the bone marrow in transplants using CD68S-GCase-targeted and CD68S-GCase-P2A-Citrine-targeted cells (blue circles: 0.25E6, green: 1E6, and red: 2E6 cells transplanted; n = 31, 33 mice). **d** Primary human engraftment in the spleen. **e** Targeted allele frequency in CD68S-GCase- and CD68S-GCase-P2A-Citrine-targeted cells before transplantation (Pre-Tx) and 16-weeks post-transplantation (Post-Tx) in engrafted human cells in the bone marrow of mice with human chimerism >1% (n = 29, 31 mice). **f** Secondary human engraftment (32 weeks) in the bone marrow (n = 8, black: CD68S-GCase-P2A-Citrine, white: CD68S-GCase). Note, three mice have chimerism <1%. **g** Targeted allele frequency before (Pre-Tx) and after transplant (Post-Tx) in the bone marrow cells of secondary mice. **a–b** Data shown as mean ± SD. **c–g** Median shown. Source data are provided as a Source Data file.

drop in modified engrafted cells relative to the pre-transplant level (43%) was 2.7-fold in the bone marrow, consistent with but in the low range of studies in NSG mice (Fig. 4e). We observed B, myeloid, and monocyte development with less preponderance of B-lymphoid population compared to NSG mice. As before, Citrine+ cells were seen exclusively in the myeloid and monocyte cells (Fig. 6c). Tissue macrophages were extracted from liver

and lung using an enzymatic method and peritoneal macrophages were obtained by analysis of peritoneal fluid. We found robust human cell populations that were CD45+ or CD45/CD11b+ as well as Citrine+ in these macrophage cell preparations (Fig. 6d–f). Samples with high cell numbers that allowed enrichment of live human-myeloid-Citrine+ for enzymatic analysis were sorted and the GCase activity measured.

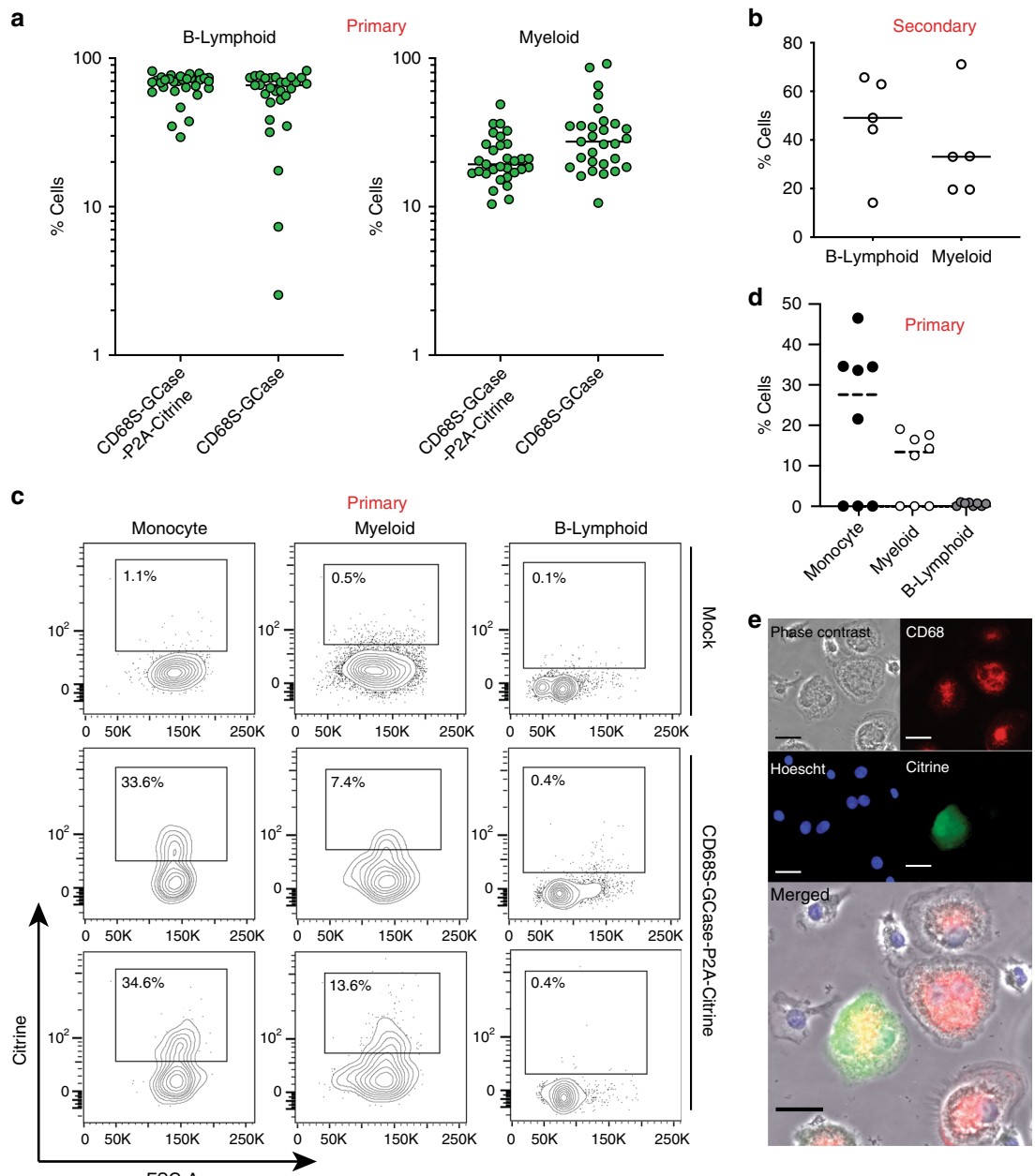

**Fig. 5 In vivo monocyte/macrophage lineage differentiation of GCase-targeted HSPCs. a** Distribution of B-lymphoid and myeloid lineage cells within the engrafted human cell population in the bone marrow from mice transplanted with CD68S-GCase and CD68S-GCase-P2A-Citrine-targeted HSPCs ($n = 29$, 31 mice). **b** Distribution of B-lymphoid and lineage cells within the engrafted human cell population from secondary transplants. Empty: vector without Citrine ($n = 5$ mice). **c** Representative FACS plots showing Citrine expression in human CD33+ (Myeloid), CD14+ (Monocyte) and CD19 (B-cells). **d** Percent Citrine-positive cells in monocyte (black), myeloid (white), and B-cell (gray) populations in mice with human CCR5 allele modification fraction >10%. **e** Representative epifluorescence microscopy images of human CD68S-GCase-P2A-Citrine-targeted macrophages differentiated from human CD14+ cells sorted from mice bone marrow and peripheral blood ($n = 10$ mice, additional examples in Supplementary Fig. 8). Images depict morphology (brightfield), nuclei (Hoechst, blue), CD68 protein (red), and Citrine (green). Scale bar is 10 μm. **a, b, d** Median shown. Source data are provided as a Source Data file.

Consistent with our studies of HSPCs differentiated in culture, the Citrine+ cells expressed 2.0 (bone marrow), 2.1 (spleen), and 1.6-fold (lungs) higher GCase than Citrine– cells (Figs. 3e and 6g). Analysis of targeted *CCR5* alleles from sorted cells populations, including bone marrow, lung, spleen, liver, and peritoneal macrophages show enrichment of targeted alleles in the Citrine+ cells compared to Citrine– cells confirming that the observed Citrine expression is from targeted cells (Fig. 6h).

## Discussion

Gaucher disease is currently treated using enzyme replacement therapy (ERT) and substrate reduction therapy (SRT). Both approaches have been shown to be effective at addressing hematological and visceral manifestations[38,39] and can reduce, but not eliminate, bone complications in this disease[40,41]. Neither ERT, not the best tolerated form of SRT (eliglustat), are expected to impact neuronopathic forms of GD (GD2 and GD3) or the increasingly recognized neurological symptoms in GD1[42,43]. ERT

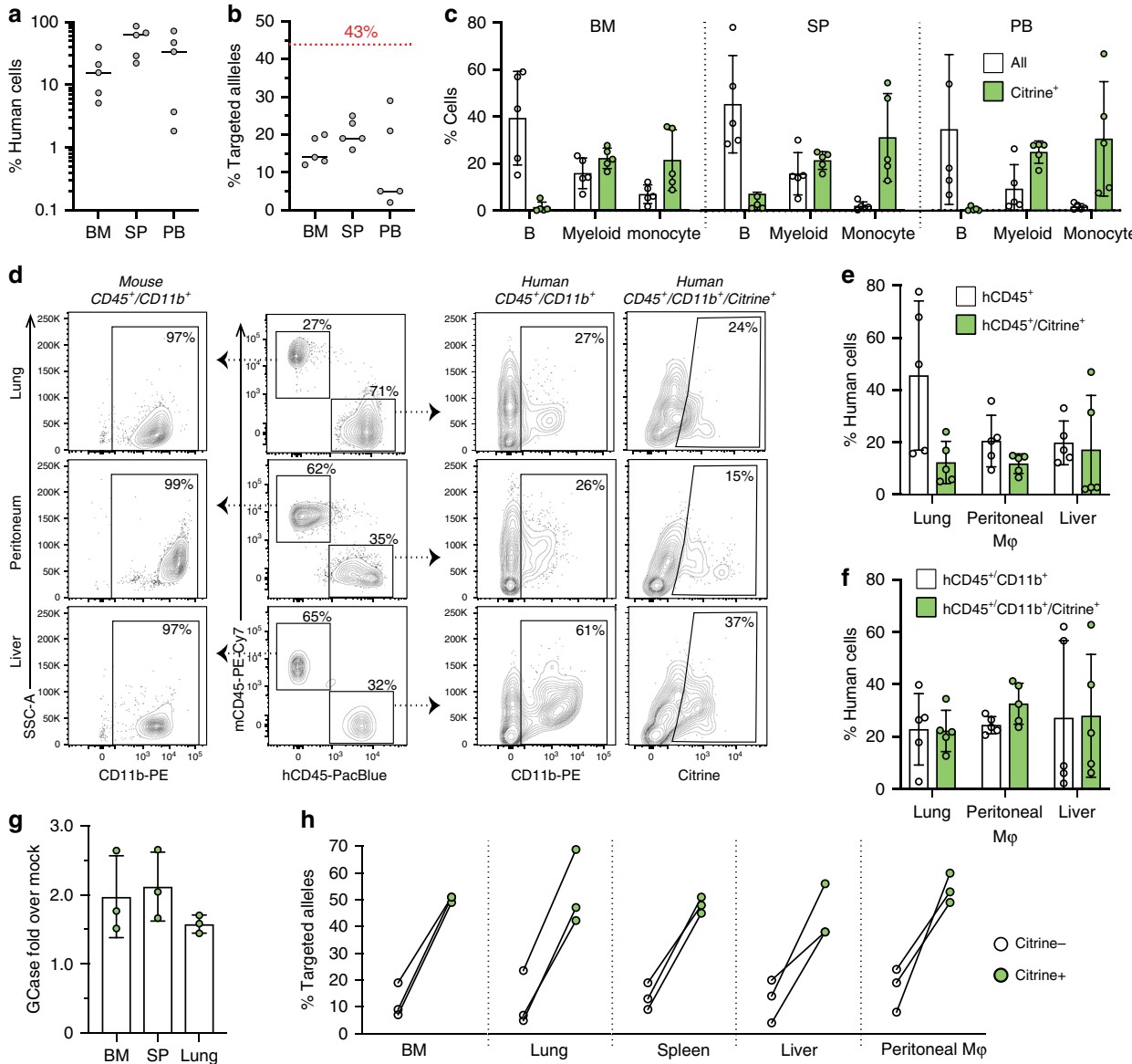

**Fig. 6 Improved macrophage differentiation of GCase-targeted HSPCs in NSG-SGM3 mice. a** Human cell engraftment 16-weeks post-transplantation in the bone marrow (BM), spleen (SP), and peripheral blood (PB) in transplants using CD68S-GCase-P2A-Citrine-targeted cells ($n = 5$ mice). **b** Modified allele frequency from engrafted CD68S-GCase-P2A-Citrine-targeted cells in the same tissues ($n = 5$ mice). **c** Percent human B-cell (CD19[+]), myeloid (CD33[+]), and monocyte (CD14[+]) populations in BM, SP, and PB shown in white. Citrine-positive cells in each population are shown in green ($n = 5$ mice). **d** Representative FACS plots showing gating strategy for mouse and human CD45[+], CD45[+]/CD11b[+], and CD45[+]/CD11b[+]/Citrine populations in macrophage preparations from lung, peritoneum, and liver. **e** Percent human CD45[+] and human CD45[+]/Citrine cells in the same preparations ($n = 5$ mice). **f** Percent human CD45[+]/CD11b[+] and human CD45[+]/CD11b[+]/Citrine cells in the same preparations ($n = 5$ mice). **g** Fold GCase activity in human Citrine[+] cells compared to human Citrine[−] cells in BM, SP, and lung ($n = 3$ mice). **h** Modified allele frequency in human Citrine[+] cells (green) compared to human Citrine[−] cells (white) in BM, SP, and lung in the same cells. **a, b** Median shown. **c, e-g** Data shown as mean ± SD. Source data are provided as a Source Data file.

involves life-long, bi-weekly infusions, and the development of antibodies can, in some cases, decrease enzyme bioavailability and impact clinical outcome[44,45]. Approved SRTs (miglustat and eli-glustat) also require life-long administration, repeated dosing (three and two times per day, respectively) and, particularly for miglustat, significant side effects due to non-specific inhibition of other enzymes[46]. Both modalities are very costly with estimated annual cost of \$300,000 to \$450,000 (estimated life-time cost of ~\$6 to \$22 million dollars) limiting their availability worldwide[47,48]. In the past, allo-HSCT was used effectively and led to rapid improvement in the hematological and visceral parameters as well as regression of skeletal disease, but given its significant morbidity and mortality, its use has been reserved for individuals

with neurologic or progressive disease unresponsive to ERT and SRT[49–52]. Specifically, allo-HSCT has shown potential to halt neurological progression in patients with GD type 3 (D3) when treated at a young age and early in the disease process[53–56].

Given the potential for HSCT to constitute a one-time therapy for GD1 and its likely beneficial effect in the central nervous system (CNS), improving the safety of HSCT for GD would be a significant development. The use of autologous HSPCs is safer because it eliminates the morbidity of graft-versus-host disease, results in faster engraftment, and can lead to earlier intervention by obviating the need for donor matching. For this reason, non-otargeted lentiviral-mediated delivery of constitutively expressed GCase is being explored in HSPCs and has yielded

promising results in murine GD models where transplantation of these cells achieved normalization of GCase levels, reduced Gaucher cell infiltration, and lowered glucocerebroside storage[16–18]. However, because of the pseudorandom integration of the viral genomes, concerns remain about its potential for tumorigenicity[19,20]. Genome editing, as a more precise genetic tool, decreases the chance of random integration and ensures more predictable and consistent transgene expression. In addition to the hematopoietic system, the liver has also been considered as potential enzyme replacement depot and in vivo liver-directed approaches using zinc finger nucleases have also been investigated in mouse models[57]. However, it is not clear the liver-secreted GCase would have the proper glycosylation to cross-correct affected cells or that it could cross into the CNS. Transplantation of ex vivo genome-edited HSPCs can provide direct replacement of pathological cells and leverages the ability of graft-derived macrophages that can migrate to the brain[14] and bone. Therefore, autologous transplantation of gene-corrected cells, if coupled with safer conditioning regimens, could be a promising therapy for GD patients regardless of disease subtype.

To begin the development of autologous transplantation of genome-edited hematopoietic stem cells, we established an efficient application of CRISPR/Cas9 to target a functional copy of GCase into human CD34+ HSPCs. Here, we use sgRNA/Cas9 and AAV6-mediated template delivery to target GCase to the *CCR5* locus, a gene previously used for the insertion and expression of therapeutic genes[24,26]. *CCR5* is considered a safe harbor because germline deletions in this gene are common (up to 10% in the Northern European population) and have no overt developmental phenotype[27]. Germline *CCR5* loss might be beneficial as it provides protection against HIV[28], and possibly smallpox[58], although it also appears to reduce protection against influenza[59] and West Nile virus[60]. Compared to genetic correction of the affected locus, the use of a safe harbor is a universal therapy for all patient mutations and has greater designability as regulatory and GCase protein sequences can be engineered with enhanced therapeutic properties. For targeting Gaucher disease specifically, it circumvents the design of genetic tools for the *GBA* locus, which can be non-specific given the presence of *GBAP*, a pseudogene with 96% sequence homology to the *GBA* gene.

To express GCase from the *CCR5* locus, we used a previously characterized derivative of the CD68 promoter and confirmed through in vitro and in vivo differentiation protocols that it achieves monocyte/macrophage-specific expression of GCase[30,31]. We reasoned that because the primary manifestations of Gaucher disease are due to pathology in monocyte/macrophage lineage cells, enzyme reconstitution in this lineage should be sufficient to provide phenotypic correction in this disease. Furthermore, our studies with the SFFV promoter did not consistently result in sustained GCase and reporter expression in human HSPCs, suggesting that high and sustained GCase in the stem and progenitor compartment might have detrimental effects. This would not be surprising, as negative impact in long-term engraftment by lysosomal enzyme overexpression has been seen previously for galactocerebrosidase[61]. Furthermore, transplantation using retrovirally transduced CD34+ HSPCs in human where GCase was driven by the LTR promoter failed to show long-term reconstitution[13]. While several reasons can explain this observation, including insufficient cell dose and lack of conditioning, one explanation is that constitutive GCase expression by the LTR had a detrimental effect in the repopulating stem cell.

We examined the ability of the targeted human HSPCs to engraft and differentiate in serial transplantation studies in immunocompromised mice and demonstrate that our approach can modify cells with long-term repopulation potential and preserves multi-lineage differentiation capacity. We re-demonstrated

a reduced repopulation capacity of the edited HSPC population in primary engraftment studies reported previously for engineered HSPCs in viral-mediated gene addition and gene-editing contexts[24,62,63]. However, the enhanced allele modification frequencies in the secondary transplants suggest that this initial decreased capacity is due to a reduced number of targeted long-term repopulating stem cells (LT-HSCs) compared to targeted shorter-lived progenitors and not to detrimental effect on engraftment per se. Interestingly, the allele targeting frequency of the engrafted cell population increased in some cases, suggesting that the variability in targeted HSPC engraftment may be accounted for by stochastic engraftment dynamics driven by oligoclonal reconstitution[64]. Even though these experiments do not achieve 100% human cell chimerism, transplantation outcomes in humans and mice indicate that low level chimerism could be sufficient to provide symptomatic relief[65,66]. Specifically, in mice, 7% wild type cell engraftment was shown to be sufficient to reverse disease pathology[67]. In our primary engraftment studies, the median allele modification frequency of the engrafted cells was ~4%, which corresponds to 4–8% of targeted cells (depending on the ratio bi-allelic or mono-allelic modification in the engrafted cells) and an 8–16% unmodified cell dose (given that our cells express twofold more GCase). Future experiments in an immunocompromised models of GD to allow engraftment and proliferation of human cells will establish the potential of these cells to correct the phenotype. Regardless of the outcome, future efforts aimed at increasing the permissiveness of long-term HSCs to undergo homology-dependent genome editing will be important for the therapeutic application of these cells.

Herein, we report the use of a genome editing to target a safe harbor to create lineage-specific expression of proteins. This approach is highly flexible and could serve as a platform to restore the expression of lysosomal enzymes and potentially other secreted proteins with therapeutic potential, provided the therapeutic cassettes are within the packaging capacity of AAV. These studies exemplify a specific use for this approach for the expression of human glucocerebrosidase as a potential intervention for the definitive treatment of GD and support further pre-clinical development of this strategy.

## Methods

**rAAV vector plasmid construction**. The CCR5 donor vectors have been constructed by PCR amplification of 500 bp left and right homology arms for the *CCR5* locus from human genomic DNA. SFFV and wild-type GBA sequences were amplified from plasmids. The CD68S sequence was obtained from Dahl et al.[68] and was cloned from a gblock Gene Fragment (IDT, San Jose, CA, USA). Primers were designed using an online assembly tool (NEBuilder, New England Biolabs, Ipswich, MA, USA) and were ordered from Integrated DNA Technologies (IDT, San Jose, CA, USA). Fragments were Gibson-assembled into a the pAAV-MCS plasmid (Agilent Technologies, Santa Clara, CA, USA). Constructs were planned, visualized, and documented using Snapgene 4.2 Software.

**rAAV production**. rAAV was produced using a dual-plasmid system as described in Khan et al.[69]. Briefly, HEK293 cells were transfected with plasmids encoding an AAV vector and AAV rep and cap genes. HEK293 cells were harvested 48-h post-transfection and lysed using three cycles of freeze-thaw. Cellular debris was pelleted by centrifugation at $1350 \times g$ for 20 min and the supernatant collected. Active rAAV particles were purified using iodixanol density gradient ultracentrifugation, dialyzed in phosphate-buffered saline (PBS), and stored in PBS at −80 °C. rAAV vectors for in vivo applications were ordered from Vigene Biosciences (Rockville, MD, USA). Viral titers were determined using droplet digital PCR with the following primer/probe combination: F: GGA ACC CCT AGT GAT GGA GTT, R: CGG CCT CAG TGA GCG A, P: /56FAM/CAC TCC CTC/ZEN/TCT GCG CGC TCG/ 3IABkFQ/.

**HSPC isolation and culturing**. Human CD34+ HSPCs mobilized from peripheral blood were purchased frozen from AllCells (Almeda, CA, USA) and thawed per manufacturer's instructions. Human Cord blood was obtained through The Binns Program for Cord Blood Research Program and not by the investigators themselves. The Program was approved by Stanford's IRB. Eligible donors were

expectant mothers scheduled to deliver at Lucile Packard Children's Hospital who provided informed consent prior to collection. Briefly, mononuclear cells were isolated by density gradient centrifugation using Ficoll Plaque Plus density gradient medium followed by two platelets washes. CD34+ mononuclear cells were positively selected using CD34+ Microbead Kit Ultrapure (Miltenyi Biotec, San Diego, CA, USA) per manufacturer's instructions. Purity of the isolation was assessed by staining cells with APC-conjugated anti-human CD34+ (Clone 561; Biolegend, San Jose, CA, USA) and analyzing the fraction of APC+ cells using an Accuri C6 flow cytometer (BD Biosciences, San Jose, CA, USA). Cells were cultured in media consisting of StemSpan SFEM II (Stemcell Technologies, Vancouver, Canada) supplemented with SCF (100 ng/ml), TPO (100 ng/ml), Flt3-Ligand (100 ng/ml), IL-6 (100 ng/ml), UM171 (35 nM), and StemRegenin1 (0.75 mM).

**Gene editing in HSPCs**. An sgRNA targeting CCR5 exon 3 (sequence; 5′-GCAGCATAGTGAGCCCAGAA-3′) was purchased from TriLink Biotechnologies (San Diego, CA, USA) with the chemical modification 2′-O-methyl-3′-phosphorothioate[25]. Cas9 and Hifi Cas9 were purchased from Integrated DNA Technologies (IDT, San Jose, CA, USA Catalog #1081058 and #1081060). The editing procedure was performed as follows: sgRNA and Cas9 protein were complexed at a molar ration of 1:2.5 (sgRNA:Cas9) at room temperature for 5 min. The RNP was electroporated into human CD34+ HSPCs 48 h after thawing using the Lonza 4D nucleofector with the following conditions: pulse code: DZ100; cell density: $1 \times 10^6$ cells in 100 μl; [Cas9]: 30 μg; [sgRNA]: 15 μg. Following electroporation, cells were immediately rescued with HSPC culture media pre-warmed to 37 °C. rAAV6 was applied to cells at a MOI of 10,000–20,000. The frequency of indel formation was quantified using Tracking Indels by Decomposition (TIDE)[70]. CCR5 expression was quantified by flow cytometry using anti-human CCR5-APC antibody (BD Biosciences, #556903).

**Measurement of cassette integration using ddPCR**. Genomic DNA was extracted from selected or unselected cell populations using QuickExtract DNA Extract Solution and digested using AFlII (New England Biosciences). Two detection probes were used in the assay to simultaneously quantify wild-type CCLR2 reference alleles gene targeted CCR5 alleles. The ratio of detected CCLR2/CCR5 events gave the fraction of targeted alleles in the original cell population. The CCR5 detection assay was designed as follows: F:5′- GGG AGG ATT GGG AAG ACA-3′, R: 5′-AGG TGT TCA GGA GAA GGA CA-3′, labeled probe: 5′- FAM/AGC AGG CAT/ZEN/GCT GGG GAT GCG GTG G/3IABkFQ-3′. The reference assay was designed as follows: F:5′-CCT CCT GGC TGA GAA AAA G-3′, R: 5′-CCT CCT GGC TGA GAA AAA G-3′, and probe: /5HEX/TGT TTC CTC/ZEN/CAG GAT AAG GCA GCT GT/3IABkFQ/. Primer and probes final concentrations were 900 and 250 nM, respectively. Twenty microliters of the PCR reaction was used for droplet generation. Forty microliters of droplets was used in a PCR reaction with the conditions: 95 °C for 10 min, 45 cycles of melting at 94 °C for 30 s, annealing at 57 °C for 30 s, and extension at 72 °C for 2 min, with a final extension at 98 °C for 10 min. All steps were performed with ramping of 2 C/s and reactions were stored at 4 °C covered from light until droplet analysis. Analysis was performed on a Qx200 Droplet Reader (Bio-Rad) detecting FAM and HEX-positive droplets. Control samples included Mock (non-modified) genomic DNA and no-template control. Data analysis was performed using Quantasoft analysis software v1.4 (Bio-Rad).

**Colony-forming unit assay and clonal genotyping**. Colony-forming Unit assays were performed using Methocult methylcellulose (StemCell Technologies) as per the manufacturer's protocol. Briefly, CD34+ HSPCs were single-sorted into 96-well flat-bottom plates (Corning) pre-filled with 100 μl Methocult. Cells were cultured for 14 days at 37 °C, 5% $O_2$ and 5% $CO_2$. Colonies were quantified and characterized morphologically by color, size, and shape as burst-forming unit—erythroid (E-BFU), colony-forming unit—erythroid (E-CFU), colony-forming unit—granulocyte/monocyte (CFU-GM) or colony-forming unit—granulocyte/erythroid/macrophage/megakaryocyte. Colonies were genotyped by extracting genomic DNA in QuickExtract DNA Extraction Reagent (Lucigen, QE09050) and performing a 3-primer in-and-out PCR to amplify both wild-type CCR5 alleles and CCR5 alleles with targeted integrations. The 3-primer in-and-out PCR utilized a forward primer out the left CCR5 homology arm (5′-CACCATGCTTGACCCA GTTT-3′), a forward primer binding the poly-adenylation signal in the cassette (5′-CGCATTGTCTGAGTAGGTGT-3′) and a reverse primer binding inside the right homology arm (5′-AGGTGTTCAGGAGAAGGACA-3′). Accupower pre-mix (Bioneer, Oakland, CA) was used for the PCR with cycling parameters: 95 °C for 5 min, and 35 cycles of 95 °C for 20 s, 72 °C for 60 s. DNA fragments were detected by agarose gel electrophoresis. Wild-type and targeted CCR5 alleles yielded bands of 590 base-pairs and 1100 base-pairs, respectively.

**Macrophage differentiation and flow cytometry**. CD34+ HSPCs were seeded at a density of $2 \times 10^5$ cells/ml in non-treated 6-well plates in differentiation medium (SFEM II supplemented with SCF (200 ng/ml), Il-3 (10 ng/ml), IL-6 (10 ng/ml), FLT3-L (50 ng/ml), M-CSF (10 ng/ml) and penicillin/streptomycin (10 U/ml)). After 48 h, non-adherent cells were removed and reseeded in a new non-treated 6-well plate at $2 \times 10^5$ cells/ml in differentiation medium. Adherent cells were

maintained in the same dish in maintenance medium (RPMI supplemented with FBS (10% v/v), M-CSF (10 ng/ml) and penicillin/streptomycin (10 U/ml)). After 2 weeks, adherent macrophages were harvested by incubation with 10 mM EDTA in PBS. For phenotypic analysis, $1 \times 10^5$ cells per condition were harvested and resuspended in 100 μl staining buffer comprises PBS supplemented with 2% FBS and 0.4% EDTA. Non-specific antibody binding was blocked (5% v/v TruStain FcX, BioLegend, #422302) and cells were stained with 2 μl of each fluorophore-conjugated monoclonal antibody (30 min, 4 °C, dark). Antibodies used were hCD34-APC (BioLegend, #343509), hCD14-BV510 (BioLegend, #301842) or hCD14-APC (Invitrogen, #17-0149-41), and hCD11b-PE (BioLegend, #101208). Propidium Iodide (1 μg/ml) was used to detect dead cells and cells were analyzed on a BD FACSAria flow cytometer.

**Phagocytosis assay**. pHrodo Red E.coli BioParticles conjugate for Phagocytosis were purchased from ThermoFisher, USA and reconstituted to 1 mg/ml in 10% FBS-containing media. Reconstituted Bioparticles were added at a final concentration of 0.1 mg/ml to IDUA-HSPC-derived macrophages and incubated at 37 °C for 1 h. The cells were then washed and bathed in imaging media (DMEM Fluorobright, 15 mM HEPES, 5% FBS). Imaging followed using the appropriate absorption and fluorescence emission maxima (560 and 585 nm, respectively) with a BZ-X710 Keyence fluorescence microscope. Images were quantified using ImageJ 1.51.

**Transplantation of CD34+ HSPCs into NSG Mice**. Targeted HSPCs (unselected) were transplanted 48 h post-targeting into sub-lethally irradiated NSG recipients. Primary transplants were performed by intrahepatic injection into newborn pups or by intrafemoral injection at 6–8 weeks of age. Approximately $1 \times 10^6$ cells were transplanted into each mouse for all primary transplants. For secondary transplants, human CD34+ HSPCs were isolated from transplanted 16-week-old-mice at the time of primary engraftment analysis using CD34+ Microbead Kit Ultrapure (Miltenyi Biotec, San Diego, CA, USA) and transplanted without pooling into a second sub-lethally irradiated NSG recipient. Secondary transplants were performed by intrahepatic injection into newborn pups.

**Assessment of human cell engraftment**. Sixteen weeks post-transplantation, peripheral blood, bone marrow and spleen were harvested from transplanted mice. The tissues were passed through 100 μm filters to achieve a single-cell suspension and red blood cells were lysed with ammonium chloride (RBC lysis buffer). Non-specific antibody staining was blocked with Trustain FX (BioLegend, #422302) for 10 min at room temperature. For primary engraftment studies cells approximately one million cells were stained with 1 μl of the following antibodies: mTer119–PE-Cy5 (Invitrogen, #15-5921-83); mCD45–PE-Cy7 (Invitrogen, #25-0453-82), and 2 μl of hCD45–PacificBlue (Biolegend, #368540); hCD19–APC (BD Biosciences, #555415); hCD33–PE (BD Biosciences, #555450); hCD14–BV711(BD Biosciences, #563373). Dead cells were detected using Blue Reactive Dye (ThermoFisher #L34961) and excluded from analysis (Supplementary Fig. 5). For secondary engraftment studies, isolated bone marrow cells were stained with the following antibodies: mTer119–PE-Cy5 (Invitrogen, #15-5921-83); mCD45–PE-Cy7 (Invitrogen, #25-0453-82); hCD45–PacificBlue (Biolegend, #368540); HLA-ABC–APC-Cy7 (Biolegend, #311426); hCD19–APC (BD Biosciences, #555415); hCD33–PE (BD Biosciences, #555450). Dead cells were detected using Propidium Iodine and excluded from study (Supplementary Fig. 7). Analysis was performed by flow cytometry on a BD FACSAria II using FACSDiva v8.0.1 software. Human engraftment was defined as the percentage of hCD45 among all (mouse or human) CD45+ cells. Analysis of all flow cytometry data was done using FlowJo v10.6.

**Glucocerebrosidase activity assay**. To facilitate comparisons between different conditions, cells were FAC-sorted prior to quantification of enzyme activity and cell number ranged from $2 \times 10^4$ to $1 \times 10^5$ cells. Protein was extracted by lysing cells in 200 μl of deionized water with a Branson Sonicator with probe, centrifuging lysates at $17,000 \times g$ for 10 min at 4 °C, and collecting the supernatant containing the soluble proteins. Protein concentration in the supernatants was measured by Bradford assay kit with BSA standard curve ranging from 0.25–0.5 mg/ml (Thermo Scientific). To prepare the GCase assay working reagent, the fluorogenic substrate 4-methylumbeliferyl-β-D-glucopyranoside (Sigma, #M3633) was dissolved to a final concentration of 5 mM in citrate/phosphate buffer (pH 5.5) supplemented with 15% (w/v) sodium taurocholate. To perform the GCase assay, 25–50 μg protein extract (50 μL) was mixed with 100 μL of working reagent and incubated for 1 h at 37 °C covered from light. Reactions were stopped with 200 μL stop buffer (0.2 M glycine/carbonate, pH 10.7). Fluorescence of 4-methylumbeliferone (4MU) liberated by GCase enzyme cleavage was measured using a Molecular Devices SpectraMax M3 multi-mode microplate reader with SoftMax Pro 7 software at excitation and emission wavelengths of 355 and 460 nm, respectively (top read). A standard curve for 4MU was established using 4MU sodium salt (Sigma) in assay buffer.

**Immunocytochemistry and imaging**. Cells were seeded on coverslips 24–48 h prior to analysis. All washes were performed with D-PBS (+calcium, +magnesium). Cells on coverslips were washed, fixed with 4% PFA in PBS for 30 min, permeabilized with 0.1% Triton-X in PBS for 10 min and blocked in 10% normal

goat serum (NGS; Gibco) containing 0.25% Triton X-100 for 30 min at 25 °C. After washing, coverslips were incubated in primary antibodies: mouse anti-CD68 (Biolegend, #333801; 1:100 dilution) and rabbit anti-GFP (Abcam, ab290; 1:500 dilution) overnight at 4 °C. Primary antibodies were thoroughly washed and coverslips were incubated with secondary antibodies (Alexa Fluor 488 donkey anti-rabbit IgG (Biolegend, #406416, and Alexa Fluor 568 goat anti-Mouse IgG (H+L) (Invitrogen/ThermoFisher, A-11004) at 1:1000 dilution for 1 h covered from light. Coverslips were washed once more and mounted on glass coverslips with mounting media containing Hoechst die. Cells were imaged on a BZ-X710 Keyence fluorescence microscope. Images were quantified using ImageJ 1.51.

**Mice.** NOD.Cg-Prkdc$^{scid}$ Il2rg$^{tm1Wjl}$/SzJ (NSG) mice were developed at The Jackson Laboratory. NOD.Cg-Prkdc$^{scid}$ Il2rg$^{tm1Wjl}$ Tg (CMV-IL3,CSF2,KITLG) 1Eav/MloySzJ were described in Wunderlich et al.[37] and Billerbeck et al.[36] and obtained from The Jackson Laboratory. Mice were housed in a 12-h dark/light cycle, temperature- and humidity-controlled environment with pressurized individually ventilated caging, sterile bedding, and unlimited access to sterile food and water in the animal barrier facility at Stanford University. All experiments were performed in accordance with National Institutes of Health institutional guidelines and were approved by the University Administrative Panel on Laboratory Animal Care (IACUC 20565 and 33365).

**Tissue macrophage isolation.** Peritoneal macrophages were isolated as single-cell suspension by injection of 6 ml of ice-cold PBS 1x in the peritoneal cavity, followed by aspiration of 4 ml of the peritoneal fluid, using syringe and 21 G needle. Liver and lung were dissected from mice after perfusion, minced and digested with 500 μg/ml Liberase TM (Roche, #05401119001) and 400 μg/ml DNase in RPMI media for 30 min at 37 °C. After incubation, tissues were passed through 100 μm filters and washed twice. Liver samples were further processed by centrifugation in 33% Percoll Plus (GE Healthcare) for 15 min at $700 \times g$, with brakes off. Red blood cells were lysed from cell pellets and a single-cell suspension was prepared. For flow cytometry, non-specific antibody binding was blocked with TruStain FcX (Biolegend, #422302) and Cd16/cd32 anti-mouse (2.4G2, BD Biosciences, #553142). Cells were stained with hCD45–PacificBlue (Biolegend, #3685340), mCD45–PE-Cy7 (Invitrogen, #25-0453-82), mTer119–PE-Cy5 (Invitrogen, #15-5921-83), and h/mCD11b-PE (BioLegend, #101208). Dead cells were detected with Blue Reactive Dye (ThermoFisher #L34961).

**Statistical analysis.** All statistical test including paired and unpaired $t$-tests, and one-way analysis of variance (ANOVA) followed by Tukey's multiple comparisons test was performed using GraphPad Prism version 7 for Mac OS X, GraphPad Software, La Jolla California USA. Data was reported as means when all conditions passed three normality tests (D'Agostino & Pearson, Shapiro–Wilk, and Kolmogorov–Smirnov (KS) normality test).

**Reporting summary.** Further information on research design is available in the Nature Research Reporting Summary linked to this article.

## Data availability

All flow cytometry datasets in this study are available in Flowrepository, experiment number FR-FCM-Z2LQ. The authors declare that the other data that support the findings of this study are present within the paper, its Supplementary Information files, or are available from the corresponding author upon reasonable request. The source data underlying Figs. 1d–e, 2b, d, 3b–f, 4a–g and 5a, b, and as well as Supplementary Figs. 1d, 3a, 4b–d, 6a–b, and 8b are provided as a Source Data file.

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

## Acknowledgements

We thank Stanford's Binns Program for Cord Blood Research for providing cells. This work was supported by the Stanford's Child Health Research Institute (CHRI), the Smart Family Parkison's disease Fund, and the National Institute of Neurological Disorders and Stroke (NINDS, P50 NS062684 for T.J.M. and 1K08NS102398-01 for N.G.-O).

## Author contributions

S.G.B. collected data, performed most experiments, carried out the analyses, and participated in manuscript preparation and figure design. E.P. collected data, performed experiments, carried out the analyses, and participated in manuscript preparation and figure design. K.L.L. assisted with mouse colony management, engraftment analysis, immunocytochemistry, and imaging. P.C. collected data and performed experiments. A.S. obtained and purified CD34+ HSPCs from donated cord blood and assisted with secondary transplants. T.J.M. participated in experimental design, provided funding and assisted with manuscript preparation. M.P.H. contributed to experimental design, and manuscript preparation. N.G.-O. conceived and directed the project, collected data, designed and cloned vectors, performed experiments and analysis with S.G.B. and E.P., and participated in manuscript preparation and figure design.

## Competing interests

M.H.P. declares that he is a consultant and has equity interest in CRISPR Tx and Allogene Tx, and he states that neither company has had input or opinions on the subject matter described in this manuscript. The other authors declare no competing interests.
