## [Peer Review File · Nature Communications]

Reviewers' comments:

Reviewer #1 (CRISPR based therapy)(Remarks to the Author):

Advances over the past ~2 decades have allowed clinic-scale, efficient targeted genetic engineering of hematopoietic stem and progenitor cells. One modality of such engineering is targeted integration into a genomic safe harbor locus of a transgene that would complement disease-causing loss-of-function mutations in a native gene. One potential limitation of this approach is potentially toxic, lineage-nonspecific expression of the transgene, which in principle could be ameliorated via the use of a cell-type-specific promoter. In the submitted manuscript, Gomes-Ospina, Porteus, and colleagues investigate whether this approach can be used in the context of preclinical experiments for a human genetic disease (Gaucher's), for which both there is both bone marrow transplant and enzyme replacement therapy evidence in support of such a strategy. This is a sensible set of studies, and the gene editing data (the area of this reviewer's primary technical expertise) support the conclusions drawn.

The authors choose the CCR5 safe harbor, which is appropriate, and use the CRISPR-Cas system in conjunction with an AAV-provided repair template, which is also appropriate since it's clinic-grade. The repair cassette contains a macrophage-specific promoter to limit expression of the transgene to the relevant lineage, which is a sensible approach. All this put together yields a robust efficiency of targeted integration as gauged by marker expression (Fig 1d) and edited allele quantitation (Fig 1e). The edited HSPCs produce macrophages that are marker-positive (Fig 2); the transgenic cassette using a lineage-specific promoter confines expression to macrophages (Fig 3). Experiments transplanting the cells into immunodeficient mice show the edited cells engraft (Fig 4e-f), albeit at a reduced efficiency, and form macrophages (Fig 5c).

Overall the data support the general conclusion (14-358) that the use of safe harbor transgenesis of a lineage-specific PTU is a relevant approach to pursue, including in the context of the development of an experimental therapeutic for GD.

Comments

One general comment: is there any evidence for AAV capture into potential off-target breaks induced by Cas9?

Fig 2b: to complement the microscopy data in Fig 2c, would it make sense to also show the FACS data in Fig 2b for citrine-CD14 double-positive cells? (eg like in Fig 5c)

7-149: should be "fluorescence"

Fig 4c and 4d: the Results section mentions a ~10 fold difference in cell dose per mouse; would it be useful to color-code the data in these two figures according to the dose used (eg higher dose of cells per given mouse would be darker grey?)

Fig 5d: it may be useful to comment on the 3 animals in the monocyte group that have robust targeted integration but are not citrine-positive.

Discussion: it may be appropriate to mention the packaging limit of AAV as one factor to consider potential disease targets for this approach.

16-409: a reference is missing

Reviewed by: Fyodor Urnov

Reviewer #2 (Gene therapy with stem cells)(Remarks to the Author):

The manuscript by Scharenberg et al. characterizes a gene therapy approach to express the enzyme glucocerebrosidase in Gaucher Disease patients. The experimental approach allows expression of the GCCase enzyme by Crispr/Cas9 editing and AAV6-mediated delivery of the gene in the CCR5 locus in human HSPCs. Several templates are tested, one of them allowing specific expression of the enzyme in the monocyte/macrophage lineage, which is then evaluated in vivo in immunodeficient humanized mice. The work is reasonably straightforward to follow, and the data are of high quality. This study provides the first example of transplantation of Crispr/Cas9-edited HSPCs expressing GCCase and are potentially important for clinical applications in the field of gene therapy and more specifically Gaucher Disease. However, there are a number of issues that need to be addressed in order to improve the clarity and accuracy/quality of the paper.

Major comments

1- Figure 1: they authors should check CCR5 expression in edited versus non-edited cells.

2- Figure 1C: Modifications of HSPCs with AAV+RNP using the SFFV construct show 2 populations of high and dim citrine+ cells but not in the CD68S construct. How do the authors explain this observation?

3- Figure 1 D and E: In figure 1D, it seems that the SFFV promoter leads to a higher expression of Citrine relative to the CD68S promoter. However, in Figure 1E, allelic integration in enriched Citrine+ cells show a slightly better integration when using the CD68S-GCCase-P2A-citrine construct in comparison with the SFFV promoter construct. Does the silencing of the CD68S promoter in the more primitive populations explain the lower citrine expression observed with bulk cells modified with CD68S promoter in Figure 1C? To support their hypothesis, the authors should clarify if these samples were also collected at 48 hours post-modification. This figure is confusing as it does not compare directly with Figure 1D. It would be good to add the data in non-enriched population in Figure 1E.

4- Figure 2B: according to the title of the paragraph (line 124) it seems that the cell phenotype was analyzed in edited human HSPCs but it would be more rigorous to compare this phenotype with non-edited cells as well.

5- Supplementary Figure 4: If the authors have looked into T cell compartment as well it would be informative to add these data.

6- Figure 4E: It is unclear what do pre- and post- mean on the x-axis. Probably pre- and post-transplant but please clarify in the figure legend or in the results section. Additionally, it is unclear in which cell population was the targeted alleles frequency determined. Indeed, this would help a more accurate evaluation of the extent of editing in this in vivo experiment. The authors should explain whether it was from the PBMCs fraction including mouse and human CD45+ cells or isolated human CD45+ cells or myeloid cells or else.

7- Figure 5B: why are only 5 mice represented when 8 mice received a secondary transplant?

8- Figure 5: It is unclear if these data are from the primary or secondary transplant (except for figure 5B). There is persistence of edited human cells and citrine expression at 16 weeks (or 32 weeks?) post editing, but it is important to confirm GCCase expression. The authors should analyze expression and function of GCCase at 32 weeks. The authors could also use the sorted cells from myeloid and B cells lineages to further confirm the specificity of CD68S promoter in vivo as well and show GCCase activity from these sorted cells relative to other lineages.

9- Figure 5C: it is confusing if these data were obtained from primary or secondary transplanted mice? If these are from primary transplant, then the same should be shown using mice from the

secondary transplants to demonstrate long term persistence of protein expression from edited cells. It is important to show expression and function of GCaase as well in both primary and secondary transplanted mice.

10- Supplementary figure 5A and B: How do the authors explain the drop in modified engrafted cells relative to the level of editing pre-transplant? The authors should comment on this observation in the results section.

Minor comments

1- Please be specific as to which in vivo model is used in the study

2- Figure 1D: the authors should clarify in the figure legend if these data were obtained by flow cytometry?

3- Figure 1E: the authors should describe how were the citrine+ versus citrine- cells enriched.

4- Figure 3A: how many days post-modifications were the cells sorted?

5- Figure 3D: how long after the sort was the GCaase activity assessed? The authors should provide some statistical analyses for these data to determine if these differences are statistically significant.

6- Figure 4B: it would be informative to know how many days post-modifications were the citrine+ / citrine- cells sorted so we have a better understanding how early in the GM differentiation phase can the CD68S promoter allow citrine expression and support the hypothesis of an expression in granulocyte/monocyte-primed progenitors.

7- From lines 239 to 243, it is written "The median percentage of myeloid cells and B-cells in the bone marrow was 24.9% and 63.6%, respectively, for the mice transplanted with CD68S-GCaase-targeted HSPCs, and 18% and 66.8%, respectively, for the mice transplanted with CD68S-GCaase-P2A-Citrine-targeted HSPCs." However, in the myeloid cells panel (Figure 5A) the % is higher for CD68S-GCaase-P2A-Citrine than CD68S-GCaase whereas the numbers indicate the opposite (18% versus 24.9%). Does Figure 5A represent the bone marrow data or else or are the myeloid numbers inverted for CD68S-GCaase and CD68S-GCaase-P2A-Citrine?

8- Figure 5A: The authors should clarify in the figure legend if these cell samples were collected from the bone marrow at necropsy? Were these samples collected at 16 weeks too? Are these mice from the primary transplant?

9- Supplementary figure 5A and B: on the x-axis, citrine is written "citrine", please correct.

10- Supplementary figure 6: It is unclear why the authors didn't use the same panel as in primary transplant to monitor the level of human cells engraftment and differentiation?

11- In the discussion (lines 346 to 353) it would be informative to discuss the study from Dunbar et al. who attempted transplant of transduced CD34+ cells in GD patients without success (reference 18) and why is it expected that the authors strategy would be more efficient.

12- In the Methods section, line 371, the origin of the CD68S sequence is missing. Similarly, line 409 the reference for 2'-O-methyl-3'-phosphorothioate is missing and line 520 the reference for the GCaase activity assay is missing. Please clarify.

Point-by-point Response to Reviewer's Comments

Responses to comments from Reviewer 1

We sincerely thank reviewer 1 for the positive comments and for acknowledging the substantial effort that went into generating this data.

Comment 1: Is there any evidence for AAV capture into potential off-target breaks induced by Cas9?

Response: *We hope to answer this question in two parts: off-target activity of our guide and specific experiments looking at off-target capture.*

1. *Analyses of off-target events using our CCR5 sgRNA has been performed previously (Gomez-Ospina, Scharenberg et al. 2019). Five off-target sites with minimal activity (<0.5%) were detected when wild-type Cas9 was used as the nuclease; these off-target cleavage events were effectively abolished using HifiCas9 (Vakulskas, Dever et al. 2018). The majority of experiments referenced in this manuscript were done with HifiCas9.*

2. *Off-target capture has been examined previously in our lab by looking at integration and long-term expression of reporter proteins in two settings: 1) mismatched guide and donor templates (Dever, Bak et al. 2016) and 2) conditions that included the AAV but not RNP (data not shown). In both of these cases the percent of cells with persistent reporter expression was ~0.1%.*

Comment 2: Fig 2b: to complement the microscopy data in Fig 2c, would it make sense to also show the FACS data in Fig 2b for citrine-CD14 double-positive cells? (e.g. like in Fig 5c)

Response: *As suggested by the reviewer, we have included the FACS data showing marker expression for the targeted cells. Figure 2 now includes graphs (2d) and representative plots (2e) showing CD11b, CD14 and CD11c/CD14 expression in all the cell populations (mock, CD68S-Citrine-, CD68S-Citrine+, SFFV-Citrine-, SFFV-Citrine+) with and without differentiation. This data demonstrates increased marker expression with differentiation for all cells tested. Citrine+ cells have higher marker expression in the undifferentiated condition (HPSC) supporting the idea that these are monocyte-committed progenitors. The data also shows that CD68S-Citrine+ cells have the highest expression of these markers in the differentiated macrophages. See lines 134-144 and Figures 2d and 2e for the full results.*

Comment 3: 7-149: should be "fluorescence"

Response: *Corrected*

Comment 4: Fig 4c and 4d: the Results section mentions a ~10-fold difference in cell dose per mouse; would it be useful to color-code the data in these two figures according to the dose used (e.g. higher dose of cells per given mouse would be darker grey?)

Response: *The dose varied for experiments done using the CD68S-GCase-P2A-Citrine but not in the CD68S-GCase experiments. Individual dots in the bone marrow and spleen engraftment as well as the allele targeting modification graphs (4d-e) have been color-coded to represent high and low cell dose. Blue dots= 0.25E6, green= 1E6, and red= 2E6 (see also update figure legend). This arrangement suggests a correlation between higher cell dose and higher targeting efficiency in the engrafted cells, but not necessarily more human engraftment. This makes sense as there might a few more targeted long-term stem cells available for engraftment. In general, based on data with other loci and targeting schemes (Dever, Bak et al. 2016, Gomez-Ospina, Scharenberg et al. 2019, Pavel-Dinu, Wiebking et al. 2019), there is a lot of human donor-to-donor variability. The results section now reads: “Because cell doses of transplantation varied in the mice targeted with the Citrine-containing construct, the mice were colored-coded and tracked for engraftment and targeting efficiency in engrafted cells. This suggested a correlation between higher cell dose and higher engraftment of modified cells, a finding that is not surprising as there are likely more targeted long-term stem cells available for engraftment.” Lines 249-253.*

Comment 5. Fig 5d: it may be useful to comment on the 3 animals in the monocyte group that have robust targeted integration but are not citrine-positive.

Response: *We can think of several reasons: The first is incomplete differentiation along this lineage, since the human cells are lacking exposure to the appropriate cytokines. The CD14+ population in NSG mice might not represent a homogeneous or well-differentiated monocyte population where the CD68 would be active. It might be possible that with time, differentiation and expression of the lineage-specific promoter would improve. Second, we used a stringent gating scheme for Citrine fluorescence to make sure unmodified cells were not included in the analysis, so perhaps we missed some low expressing cells. Finally, given the stochastic and clonal nature of engraftment and repopulation by edited human HSPCs in the NSG model, it is possible that allelic modification can vary across the different lineages. For the transplant experiments described in figure 5, allelic modification was analyzed on bulk bone marrow cells but not for each lineage (monocyte, myeloid and B-lymphoid subpopulations). This is an interesting question and one that we are looking at with ongoing experiments but are no longer able to query with these samples. The following line was added to the results: “Despite robust modification in the bone marrow, three mice did not show Citrine expression in monocytes, which could be due to incomplete differentiation along this lineage since the human cells are lacking the appropriate cytokines or expression that is below our stringent gating strategy” Lines 290-293.*

Comment 6. Discussion: it may be appropriate to mention the packaging limit of AAV as one factor to consider potential disease targets for this approach.

Response: *We have added the last paragraph in the discussion where we address generatability of our approach to read: “This approach is highly flexible and could serve as a platform to restore the expression of lysosomal enzymes and potentially other secreted proteins with therapeutic potential, provided the therapeutic cassettes are within the packaging capacity of AAV (lines 434-435)*

Comment 7. 16-409: a reference is missing

Response: *Corrected*

Reviewed by: Fyodor Urnov

Responses to comments from Reviewer 2

We thank reviewer 2 for the thorough assessment of our manuscript. Based on the comments we have substantially revised the manuscript. Most significantly, we added additional experiments, including a 16-week transplantation study aimed primarily at measuring long-term GCa6 expression from our modified human cells in mice. We also performed experiments in culture to examine CCR5 expression as requested by the reviewer. In addition, we have made many modifications to the text and figures in order to improve readability of our manuscript and performed an independent re-analysis of all our data to confirm reproducibility of the numbers being reported.

Major comments

Comment 1. Figure 1: the authors should check CCR5 expression in edited versus non-edited cells.

Response: *We examined CCR5 expression in differentiated cells and have included this data as Supplementary Figure 3. The results now read (line 145-159): “CCR5 is absent from HSPCs but becomes expressed with monocyte/macrophage differentiation. To examine the effect of our genome editing process on CCR5 expression we targeted human HSPCs, differentiated them, and quantified CCR5 protein by FACS (Supplementary figure 3). In the RNP alone condition, the efficiency of double-strand DNA break generation by our CCR5 RNP complex was estimated by measuring the frequency of insertions/deletions (Indel) at the predicted cut site. The mean indel frequencies in the undifferentiated and differentiated populations was $96.8\% \pm 1.2$ and $96.4\% \pm 1.6$ respectively, resulting in almost complete knock-down of CCR5 protein expression (Supplementary figure 3a). In the presence of both RNP and AAV, cells that successfully underwent HDR (Citrine +) generally lack CCR5 expression, consistent with disruption of both CCR5 alleles by either bi-allelic integration of the cassette or mono-allelic with indel formation in the second allele (Supplementary figure 3b). In the presence of AAV, CCR5+ cells can be found in the population that did not undergo HDR (~20%), suggesting that AAV transduction decreases indel generation or exerts a small negative selection in cells containing AAV and RNP.”*

Comment 2. Figure 1C: Modifications of HSPCs with AAV+RNP using the SFFV construct show 2 populations of high and dim citrine+ cells but not in the CD68S construct. How do the authors explain this observation?

Response: *We and others have studied the dynamics of reporter protein expression following genome editing with RNP + AAV (Dever, Bak et al. 2016, Gomez-Ospina, Scharenberg et al. 2019). Two to three days after genome editing targeted cells with “high” and “intermediate” fluorescence can be observed in all loci tested. Cells with intermediate fluorescence can be a mixture of cells showing episomal expression of the AAV, cells that have been targeted but are just starting to ramp protein expression, or specific cell types that have intrinsically lower levels of transgene expression (as HSPCs are a heterogenous population). In general, with time most cells coalesce into a single more uniform*

population, as can be seen in Figure 2a. Supplementary Figure 4 further examines the long-term behavior of the SFFV-GCase-P2A-Citrine targeted HSPCs in culture, as it is unusual in that with other constructs cells usually coalesce into the high population, but not for this construct. The allele modification frequency does not change, which argues against negative selection of modified cells, so we think this is due to transcriptional and/or translational regulation of the SFFV promoter. We discuss this in detail in the results section (lines 173-179).

Given the low levels of transcription from the CD68 promoter in HSPCs it is hard to discern any subpopulations, but it is likely to be more uniform as only certain cells should have this active promoter, and episomal expression from the AAV is minimal.

Comment 3. Figure 1 D and E: In figure 1D, it seems that the SFFV promoter leads to a higher expression of Citrine relative to the CD68S promoter. However, in Figure 1E, allelic integration in enriched Citrine+ cells show a slightly better integration when using the CD68S-GCase-P2A-citrine construct in comparison with the SFFV promoter construct. Does the silencing of the CD68S promoter in the more primitive populations explain the lower citrine expression observed with bulk cells modified with CD68S promoter in Figure 1C? To support their hypothesis, the authors should clarify if these samples were also collected at 48 hours post-modification. This figure is confusing as it does not compare directly with Figure 1D. It would be good to add the data in non-enriched population in Figure 1E.

Response: We added the % targeted alleles for the non-enriched populations (bulk) to Figure 1e as the reviewer suggested and we have changed the Figure legend and the main text for clarification (See lines 92-124). All studies in Figure 1 are done 48-hours post-modification. Panel 1d shows the percent Citrine+ in the bulk preparations. Panel 1e shows the % targeted alleles in AAV only, bulk cells (corresponding to 1d), FAC-sorted Citrine+ and Citrine- cells. The expectation is that for CD34+ hematopoietic stem and progenitor cells (HSPCs) in an undifferentiated state there would be low expression from the CD68 promoter. Even if the number of modified alleles is significant, the promoter is not yet active. CD34+ HSPCs are a mixture of true stem cells and progenitors, so the low levels of expression observed is likely due to the presence of progenitors already committed to a monocyte/macrophage lineage. Without differentiation and activation of the CD68 promoter it is not really possible to compare the SFFV and CD68 constructs in terms of reporter or GCase expression.

Comment 4. Figure 2B: according to the title of the paragraph (line 124) it seems that the cell phenotype was analyzed in edited human HSPCs but it would be more rigorous to compare this phenotype with non-edited cells as well.

Response: In all of our experiments edited cells are compared to Mock-treated cells. These are cells that undergo electroporation but are not exposed to RNP or AAV. Figures 2a and 2b show the differentiation protocol in Mock-treated cells that are maintained in either HSPC media (CD34+ HSPCs) or in differentiation media (HSPC-macrophages) and compare the markers with macrophages derived from human monocytes obtained from adult peripheral blood. The goal here was to validate the differentiation with our control cell population (Mock).

We have redacted the results sections to make sure this is clear and have added two new panels 2d and 2e that show the phenotype in all cell populations, including mock, CD68-GCase-P2A targeted (Citrine+

and Citrine- populations), and SFFV-GCase-P2A targeted (Citrine+ and Citrine- populations). See revised result section (lines 128-144).

The title is meant to convey the main point of this section: That our edited cells can in fact make functional macrophages, although we first described the protocol in unedited cells.

Comment 5. Supplementary Figure 4: If the authors have looked into T cell compartment as well it would be informative to add these data.

Response: We agree that this data would be very informative. NSG mice are notoriously poor at supporting T-cell development, and experiments by us (Gomez-Ospina, Scharenberg et al. 2019) and others have shown very low numbers of undifferentiated “T-cells” (as judged by CD3 positivity). This is partly due to the fact that the murine thymic microenvironment can only support some aspects of human T-cell development. Based on this we did not look in the T-cell compartment.

Comment 6. Figure 4E: It is unclear what do pre- and post- mean on the x-axis. Probably pre- and post-transplant but please clarify in the figure legend or in the results section. Additionally, it is unclear in which cell population was the targeted alleles frequency determined. Indeed, this would help a more accurate evaluation of the extent of editing in this in vivo experiment. The authors should explain whether it was from the PBMCs fraction including mouse and human CD45+ cells or isolated human CD45+ cells or myeloid cells or else.

Response: We apologize for the confusion. We have modified the figure legend and results section (see lines 239-243). Targeted allele frequency was measured in the bulk cell population before transplantation and in the human cells engrafted in the bone marrow. This is measured using droplet digital PCR targeting the edited human allele normalized by a reference probe to a neighboring gene (CCRL2). Because the assay targets only human alleles it does not matter whether we use the PBMC fraction that includes mouse and human CD45+ cells or sorted human CD45+ cells. The answer is the same, though with enriched human cells it is less noisy. Generally, we used the PBMC fraction unless human cell engraftment was too low, in which case we enriched for human cells by FACS. See also new result section (Figure 6) and response to comment 8, where we used a different mouse line to enhance macrophage differentiation and measured targeted CCR5 alleles in sorted human cell populations.

Comment 7. Figure 5B: why are only 5 mice represented when 8 mice received a secondary transplant?

Response: Mice with low human cell chimerism (<1%), have low cells numbers which often prevent the reliable quantitation of targeted human alleles and human subpopulations. Of the 8 mice in the secondaries transplants 3 had engraftment that was less than 1%. The results section and figure legends now say this explicitly (lines 273-275). Thank you for pointing this out.

Comment 8. Figure 5: It is unclear if these data are from the primary or secondary transplant (except for figure 5B). There is persistence of edited human cells and citrine expression at 16 weeks (or 32 weeks?) post editing, but it is important to confirm GCase expression. The authors should analyze expression and function of GCase at 32 weeks. The authors could also use the sorted cells from

myeloid and B cells lineages to further confirm the specificity of CD68S promoter in vivo as well and show GCCase activity from these sorted cells relative to other lineages.

Response: *Figure 5 combines primary and secondary transplants to examine the question of lineage differentiation in mice and we agree that it is confusing. We have added more details to the text and figure legend to help clarify. In this figure we looked at Citrine expression as a surrogate for GCCase expression in primary transplants that are 16-weeks post-engraftment (panels 5c-e).*

Because enzyme activity cannot differentiate between mouse or human cells and expression from our cassettes is not high enough to be detected in mixed human cell populations, to look at GCCase activity in modified cells after engraftment we needed live mice transplanted with the Citrine containing construct (CD68S-GCCase-P2A-Citrine) so targeted cells (Citrine+) could be isolated and assayed for enzymatic activity and allele modification. To do this, we performed additional transplantation studies. To enhance differentiation of our human cells we used a different mouse line that combines features of the highly immunodeficient NOD scid gamma (NSG) mouse (used in our prior experiments) with human cytokines that support the stable engraftment of myeloid lineages (IL-3, GM-CSF, and SCF) (Figure 6). Briefly, analysis 16-weeks post-engraftment in these mice showed improved macrophage differentiation in multiple tissues and Citrine expression that is again restricted to myeloid and monocyte lineages. FACS-based isolation of human/myeloid/Citrine+ cells re-demonstrated ~2-fold more GCCase activity in Citrine+ cells compared to Citrine- cells, similar to what was found in culture. Furthermore, examination of targeted alleles in Citrine+ and Citrine- population from various tissues confirmed genetic modification of the Citrine+ cells. We think this is very strong data supporting the engraftment and differentiation of our human edited cells and the appropriate and long-term expression of the GCCase-P2A-cassette. For a full description of the results see lines 305-331 and Fig 6.

The reviewer suggested that GCCase expression should be confirmed at 32 weeks. We performed secondary transplants for the purposes of looking at long-term repopulation of our edited cells and to answer two basic questions: are the cells there? are they still modified?

For this experiment, we selected 8 mice with high primary engraftment to maximize the number of cells available for transplantation. Of these, 7 came from mice transplanted with the donor lacking citrine (CD68S-GCCase, shown as open circles) so we are unable to appropriately look at enzyme activity in these mice. For the one secondary mouse transplanted with cells modified with the citrine-containing donor, the engraftment was too low to perform a reliable assessment. Citrine expression was observed in the myeloid lineage for that mouse at low levels but given the low number of cells and the absence of replicates the data was not reported.

We think that a solid demonstration of enzyme and reporter expression 16-weeks post-transplantation is sufficient evidence to support long-term expression of the cassette. We cannot think of a biological mechanism by which the cassette would be silenced by re-transplantation of the stem cells given that silencing was not observed with sustained engraftment, massive expansion and differentiation for 4 months. Starting such an experiment now would take ~9 months.

Comment 9. Figure 5C: it is confusing if these data were obtained from primary or secondary transplanted mice? If these are from primary transplant, then the same should be shown using mice

from the secondary transplants to demonstrate long term persistence of protein expression from edited cells. It is important to show expression and function of GCCase as well in both primary and secondary transplanted mice.

Response: See response to comment 8.

Comment 10. - Supplementary figure 5A and B: How do the authors explain the drop in modified engrafted cells relative to the level of editing pre-transplant? The authors should comment on this observation in the results section.

Response: *The drop in allele modification fraction has been seen by us and others and it is important enough that we address it in the discussion. Paragraph #5 of the discussion section reads: "We examined the ability of the targeted human HSPCs to engraft and differentiate in serial transplantation studies in immunocompromised mice and demonstrate that our approach can modify cells with long-term repopulation potential and preserves multi-lineage differentiation capacity. We re-demonstrated a reduced repopulation capacity of the edited HSPC population in primary engraftment studies reported previously for engineered HSPCs in viral-mediated gene addition and gene-editing contexts^{29,30,76}. However, the enhanced allele modification frequencies in the secondary transplants suggest that this initial decreased capacity is due to a reduced number of targeted long-term repopulating stem cells (LT-HSCs) compared to targeted shorter-lived progenitors and not to detrimental effect on engraftment per se. ..."*

Minor comments

1- Please be specific as to which in vivo model is used in the study

Response: *Lines 224-226 in results lines 624-627 in the methods describes the NSG mouse line.*

2- Figure 1D: the authors should clarify in the figure legend if these data were obtained by flow cytometry?

Response: *The legend now reads: flow cytometric quantification of Citrine+....*

3- Figure 1E: the authors should describe how were the citrine+ versus citrine- cells enriched.

Response: *This figure legend has been modified to say FACS-enriched and includes additional analyses to address comment 3.*

4- Figure 3A: how many days post-modifications were the cells sorted?

Response: *Cells were sorted 48-hours post-modification. This was added to the figure legend.*

5- Figure 3D: how long after the sort was the GCCase activity assessed? The authors should provide some statistical analyses for these data to determine if these differences are statistically significant.

Response: *GCCase activity was measured after sorting without additional culturing. The figure legend has been modified and statistics added.*

6- Figure 4B: it would be informative to know how many days post-modifications were the citrine+ / 01citrine- cells sorted so we have a better understanding how early in the GM differentiation phase can the CD68S promoter allow citrine expression and support the hypothesis of an expression in

granulocyte/monocyte-primed progenitors.

Response: *Cells were sorted 48-hours post-modification, but morphology assessment in colony formation assays is done after 2 weeks (See methods, Colony-Forming Unit Assay and Clonal Genotyping, lines 513-532). We modified the results section to include this see lines 210-212.*

7- From lines 239 to 243, it is written “The median percentage of myeloid cells and B-cells in the bone marrow was 24.9% and 63.6%, respectively, for the mice transplanted with CD68S-GCase-targeted HSPCs, and 18% and 66.8%, respectively, for the mice transplanted with CD68S-GCase-P2A-Citrine-targeted HSPCs.” However, in the myeloid cells panel (Figure 5A) the % is higher for CD68S-GCase-P2A-Citrine than CD68S-GCase whereas the numbers indicate the opposite (18% versus 24.9%). Does Figure 5A represent the bone marrow data or else or are the myeloid numbers inverted for CD68S-GCase and CD68S-GCase-P2A-Citrine?

Response: *Thanks for pointing this out. To be cautious, we reviewed of all our FACS data. The numbers did not change significantly. The labels were in fact inverted in the figure and the mistake was corrected.*

8- Figure 5A: The authors should clarify in the figure legend if these cell samples were collected from the bone marrow at necropsy? Were these samples collected at 16 weeks too? Are these mice from the primary transplant?

Response: *Yes. See responses to major comment 8. The labels, figure legend and text that describe figure 5 have been changed for clarity.*

9- Supplementary figure 5A and B: on the x-axis, citrine is written “cirine”, please correct.

Response: *Thank you. This has been corrected.*

10- Supplementary figure 6: It is unclear why the authors didn't use the same panel as in primary transplant to monitor the level of human cells engraftment and differentiation?

Response: *There is no specific reason why we used a different panel. We performed secondary transplants for the purposes of looking at long-term repopulation of our edited cells to and answer two basic questions: are the cells there and are they still modified? We used a panel used extensively in the lab and that is more applicable for most of our FACS machines as it does not need a UV laser.*

11- In the discussion (lines 346 to 353) it would be informative to discuss the study from Dunbar et al. who attempted transplant of transduced CD34+ cells in GD patients without success (reference 18) and why is it expected that the authors strategy would be more efficient.

Response: *Thank you for pointing this reference. We can think of a few reasons why the authors did not see sustained engraftment of these cells but there are probably more. The first one is that engraftment in humans (and mice) requires conditioning, as this opens up stem cell niches in the bone marrow that can be occupied by the transfused HSCs. It is also possible that the cells dose was too low. More relevant to us is that the authors used a retroviral vector that expressed GCase under the control of LTR*

promoter, which is constitutive and may detrimentally affect the stem cells. This would support our hypothesis that restricting expression of GCCase outside of the stem cell compartment would likely be beneficial. We have incorporated the following in paragraph #4 of our discussion: "Furthermore, transplantation using retrovirally transduced CD34+ HPSCs in humans where GCCase was driven by the LTR promoter failed to show long-term reconstitution (Dunbar, Kohn et al. 1998). While several reasons can explain this observation, including cell dose and lack of conditioning, one explanation is that constitutive GCCase expression by the LTR had a detrimental effect in the repopulating stem cell." Lines 400-405.

12- In the Methods section, line 371, the origin if the CD68S sequence is missing. Similarly, line 409 the reference for 2'-O-methyl-3'-phosphorothioate is missing and line 520 the reference for the GCCase activity assay is missing. Please clarify.

Response: *Thanks again. The missing references have been added.*

References

1. Dever, D. P., R. O. Bak, A. Reinisch, J. Camarena, G. Washington, C. E. Nicolas, M. Pavel-Dinu, N. Saxena, A. B. Wilkens, S. Mantri, N. Uchida, A. Hendel, A. Narla, R. Majeti, K. I. Weinberg and M. H. Porteus (2016). "CRISPR/Cas9 beta-globin gene targeting in human haematopoietic stem cells." Nature **539**(7629): 384-389.
2. Dunbar, C. E., D. B. Kohn, R. Schiffmann, N. W. Barton, J. A. Nolta, J. A. Esplin, M. Pensiero, Z. Long, C. Lockey, R. V. Emmons, S. Csik, S. Leitman, C. B. Krebs, C. Carter, R. O. Brady and S. Karlsson (1998). "Retroviral transfer of the glucocerebrosidase gene into CD34+ cells from patients with Gaucher disease: in vivo detection of transduced cells without myeloablation." Hum Gene Ther **9**(17): 2629-2640.
3. Gomez-Ospina, N., S. G. Scharenberg, N. Mostrel, R. O. Bak, S. Mantri, R. M. Quadros, C. B. Gurusurthy, C. Lee, G. Bao, C. J. Suarez, S. Khan, K. Sawamoto, S. Tomatsu, N. Raj, L. D. Attardi, L. Aurelian and M. H. Porteus (2019). "Human genome-edited hematopoietic stem cells phenotypically correct Mucopolysaccharidosis type I." Nat Commun **10**(1): 4045.
4. Naldini, L. (2015). "Gene therapy returns to centre stage." Nature **526**(7573): 351-360.
5. Pavel-Dinu, M., V. Wiebking, B. T. Dejene, W. Srifa, S. Mantri, C. E. Nicolas, C. Lee, G. Bao, E. J. Kildebeck, N. Punjya, C. Sindhu, M. A. Inlay, N. Saxena, S. S. DeRavin, H. Malech, M. G. Roncarolo, K. I. Weinberg and M. H. Porteus (2019). "Gene correction for SCID-X1 in long-term hematopoietic stem cells." Nat Commun **10**(1): 1634.
6. Vakulskas, C. A., D. P. Dever, G. R. Rettig, R. Turk, A. M. Jacobi, M. A. Collingwood, N. M. Bode, M. S. McNeill, S. Yan, J. Camarena, C. M. Lee, S. H. Park, V. Wiebking, R. O. Bak, N. Gomez-Ospina, M. Pavel-Dinu, W. Sun, G. Bao, M. H. Porteus and M. A. Behlke (2018). "A high-fidelity Cas9 mutant delivered as a ribonucleoprotein complex enables efficient gene editing in human hematopoietic stem and progenitor cells." Nat Med **24**(8): 1216-1224.

Reviewer #2 (Remarks to the Author):

the authors have addressed all of my comments / critiques

Point-by-point Response to Reviewer's Comments

Responses to comments from Reviewer 1

We did not receive any additional comments from Reviewer 1.

Responses to comments from Reviewer 2

Reviewer #2 (Remarks to the Author):

the authors have addressed all of my comments / critiques

We sincerely thank the reviewers for their time and effort and to improve and clarify our manuscript.